# Spatial mapping of DNA synthesis reveals dynamics and geometry of human replication nanostructures

Michael Hawgood [ID][1,2,5], Bruno Urién [ID][1,2,5], Ana Agostinho [ID][2,3], Praghadhesh Thiagarajan [ID][1,2], Giovanni Giglio[1,2], Yiqiu Yang [ID][1,2], Xue Zhang [ID][1,2], Gemma Quijada[1,2], Matilde Fonseca [ID][1,2], Jiri Bartek[1,2,4], Hans Blom[2,3] & Bennie Lemmens [ID][1,2✉]

## Abstract

DNA replication is essential to life and ensures the accurate transmission of genetic information, which is significantly disturbed during cancer development and chemotherapy. While DNA replication is tightly controlled in time and space, methods to visualise and quantify replication dynamics within 3D human cells are lacking. Here, we introduce 3D-Spatial Assay for Replication Kinetics (3D-SPARK), an approach enabling nanoscale analysis of DNA synthesis dynamics in situ. 3D-SPARK integrates optimised nucleotide analogue pulse labelling with super-resolution microscopy to detect, classify, and quantify replication nanostructures in single cells. By combining immunofluorescence techniques with click chemistry-based nascent DNA labelling and transfection of fluorescent nucleotide derivatives, we map multi-colour DNA synthesis events in relation to established replication proteins, local RNA-protein condensates or large subnuclear domains. We demonstrate quantitative changes in size, relative abundance and spatial arrangement of nanoscale DNA synthesis events upon chemotherapeutic treatment, CDC6 oncogene expression and loss of chromatin organiser RIF1. The flexibility, precision and modular design of 3D-SPARK helps bridging the gap between spatial cell biology, genomics, and 2D fibre-based replication studies in health and disease.

**Keywords** DNA Replication Dynamics; Nascent DNA Labelling; Nanoscale Imaging; Super-Resolution Microscopy; Genome Architecture
**Subject Categories** Chromatin, Transcription & Genomics; DNA Replication, Recombination & Repair; Methods & Resources

## Introduction

DNA replication is a fundamental biological process, essential for cellular proliferation, genome stability and overall organismal health. It ensures the complete and accurate duplication of the genome precisely once per cell cycle and follows a defined temporal and spatial order known as the replication timing (RT) programme. This programme is highly conserved among vertebrate species (Masai and Foiani, 2017) and causes the characteristic replication foci patterns observed in early, mid and late S-phase cells (Leonhardt et al, 2000; Chagin et al, 2016). While the dimensions and relative timings of individual replication sites strikingly parallel the 3D architecture of chromatin and its organisation into topologically associated domains, the functional relationships between DNA replication kinetics, chromosome organisation and cellular signalling events remain elusive (Masai and Foiani, 2017). The physiological importance of these nanoscale replication events, however, is well established, as they directly influence mutation rates and cell fate in normal and pathological conditions (Maya-Mendoza et al, 2018; Nakatani et al, 2022). DNA replication stress and alterations in RT are common features of cancer cells and cause genomic as well as epigenetic instability (Halazonetis et al, 2008; Rhind and Gilbert, 2013; Wei Dai, 2014; Ferguson et al, 2015; Weiner et al, 2024; Dietzen et al, 2024; Hosea et al, 2024). Understanding the dynamics of DNA replication—how and when replication occurs within the 3D nuclear environment—is thus central to advancing our knowledge of one of the core principles of life, which impacts normal development and disease.

Pioneering work in the 1980s revealed that DNA replication occurs in discrete, microscopically visible sites within the nucleus, which later were termed replication foci and laid the foundation for studies on RT and genome architecture (Celis and Celis, 1985; Nakamura et al, 1986). How researchers detect and measure DNA replication dynamics in human cells has evolved significantly over time, moving from radioactive nucleotides and bulk-cell measurements to current click-functionalised nucleotides and high-precision fluorescence microscopy or DNA sequencing methods. While replication foci have been recognised as fundamental units of 3D genome organisation (Jackson and Pombo, 1998; Pope and Gilbert J. Mol. Biol. 2013; Olivares-Chauvet et al, 2011), many questions remain regarding the kinetics of DNA synthesis in subnuclear space. Techniques such as DNA fibre assays and nascent DNA sequencing have provided valuable insights into replication dynamics, but lack spatial context. Alternative methods such as high-content imaging, flow cytometry and single-cell DNA

[1]Department of Medical Biochemistry and Biophysics, Karolinska Institutet, Stockholm, Sweden. [2]Science for Life Laboratory, Solna, Sweden. [3]Department of Applied Physics, Royal Institute of Technology, Solna, Sweden. [4]Danish Cancer Society Research Centre, Copenhagen, Denmark. [5]These authors contributed equally: Michael Hawgood, Bruno Urién. ✉E-mail: bennie.lemmens@ki.se

sequencing address some of these challenges but currently lack the spatial and temporal resolution necessary to resolve individual DNA synthesis events. Moreover, these approaches do not directly relate local replication dynamics to specific proteins or nuclear structures.

Super-resolution imaging techniques, such as 3D-structured illumination microscopy (3D-SIM), allow the detection of individual replicons in mammalian cells (Chagin et al, 2016). Super-resolution microscopy studies predominantly use a single nascent DNA marker such as EdU or BrdU to visualise replication forks and correlate these sites with the spatial distribution of replication factors and DNA repair proteins through multiplexed immunostaining or proximity assays. This approach has enabled precise mapping of protein recruitment to replication nanofoci under both unperturbed and stressed conditions, providing critical insights into fork stability, replication stress responses, and genome maintenance mechanisms (Lee et al, 2021; Pradhan et al, 2024; Su et al, 2020; Triemer et al, 2018; Whelan et al, 2020; Whelan and Rothenberg, 2021). Yet, the use of fluorescence microscopy to define 3D DNA synthesis dynamics at nanoscale resolution remains limited due to practical challenges in nascent DNA labelling. Common methods to introduce labelled DNA into cells, such as cell scratching, micro-injection, or electroporation, are invasive, inefficient, and hard to combine with short nucleotide pulse timings. The use of multiple nascent DNA markers—a common practice for DNA fibre techniques—would significantly benefit in situ replication analyses, but developing bioorthogonal, multi-colour pulse labelling schemes compatible with cellular super-resolution microscopy remains a challenge. Nucleotide analogues such as BrdU or AzC, for example, can yield replication foci at the nanometre scale, but require harsh DNA denaturation protocols for detection or cause widespread stalling of replication forks, respectively (Chagin et al, 2016; Triemer, et al, 2018).

Here, we set out to address these challenges and develop a flexible methodology that allows researchers to detect, classify and quantify elemental DNA replication nanostructures in single human cells, and assess DNA replication dynamics in its native 3D environment. We optimised and combined nucleotide analogue pulse labelling protocols to develop an effective multi-colour nascent DNA labelling strategy for 3D-SIM and expansion microscopy (ExM). This integrated methodology, coined 3D-spatial assay for replication kinetics (3D-SPARK), allowed us to directly link DNA synthesis dynamics and nascent DNA geometry to key regulators of genome architecture, common pathological events and nuclear protein structures. Additionally, we integrated 3D-SPARK with immunofluorescence to examine DNA replication stress and demonstrate alterations in size, abundance and spatial arrangement of nanoscale DNA synthesis events upon drug treatments and different genetic contexts.

## Results

### Optimised nascent DNA labelling for nanoscale imaging

To study DNA replication dynamics in the 3D human genome, we first optimised fluorescent labelling of newly synthesised DNA for detection by super-resolution microscopy. We aimed to generate fluorescent nascent DNA tracks that contain enough labelled

nucleotides for accurate detection, are sufficiently short to resolve single replicons, and represent effective DNA synthesis rather than fork stalling events. Classical replication kinetics assays using 2D DNA fibres rely on thymidine analogues such as BrdU, CldU or IdU that, for detection, require denaturation of the DNA helix structure and bulky antibodies that show significant cross-reactivity due to the high similarity among these nucleoside analogues (Manders et al, 1992; Salic and Mitchison, 2008; Tuttle et al, 2010; Liboska et al, 2012). To establish 3D-SPARK, we thus focused on two alternative approaches that do not require DNA denaturation and label nascent DNA directly with small fluorescent dyes (Fig. 1A).

The first approach is based on alkyne-modified nucleotides such as EdU (5-ethynyl-2-deoxyuridine) that allows attachment of fluorescent dyes via a Cu(I)-catalysed azide-alkyne cycloaddition (CuAAC) 'click' reaction (Kolb et al, 2001; Fantoni et al, 2021). This click chemistry-based labelling approach is simple, selective and does not require DNA denaturation, allowing detection of nascent DNA with a wide range of dyes and with a superior signal-to-noise ratio compared to antibody-based methods (Flomerfelt and Gress, 2016). Additionally, the small size and neutral charge of EdU allow for efficient penetration into cells, resulting in consistent labelling of DNA replication among individual cells (Fig. 1B, lower left panel). The second approach is based on immediate transfection of pre-labelled, fluorescent deoxyuridine triphosphate (dUTP) derivatives into cells by a recently developed synthetic nucleoside triphosphate transporter (SNTT1) (Zawada et al, 2018). Transported nucleotide triphosphates can be used directly for DNA synthesis without requiring enzymatic conversion, allowing for short and well-controlled labelling times (Fig. 1A) (Kužmová et al, 2021). The flexibility and complementarity of these labelling approaches make them an effective basis for quantitative multi-colour nanoscopy.

To optimise the click-chemistry-based approach, we directly compared two clickable thymidine analogues, EdU and F-ara-EdU by assessing their incorporation efficiency across a range of concentrations and pulse durations in human RPE1 cells. While both analogues showed increased nuclear signal intensity with time and concentration, EdU showed superior sensitivity at lower concentrations and shorter pulse durations. For instance, a 15-min pulse with $10\,\mu M$ EdU yielded a mean nuclear signal intensity comparable to a 4-h pulse with $100\,\mu M$ F-ara-EdU (Figs. 1B and EV1A). These observations suggest that EdU incorporation into genomic DNA is ~160-fold more efficient compared to F-ara-EdU, which makes EdU the preferred choice for detecting relatively short nucleotide pulses or replication nanostructures.

To determine the optimal conditions for the SNTT1-based approach, we transfected human RPE1 cells with three different fluorescent nucleotide triphosphates, i.e. Cy3-xx-dUTP, Atto-488-xx-dUTP or Cy3-dCTP, and compared DNA replication labelling efficiencies in single cells using quantitative image-based cytometry (QIBC). While all three nucleotide analogues incorporated into replicating nuclei, the average nuclear intensity of Cy3-xx-dUTP was two to eightfold higher compared to Cy3-dCTP or Atto-488-xx-dUTP, respectively (Fig. 1C). Plotting Cy3-xx-dUTP versus DAPI intensity in individual nuclei revealed a well-defined arc-shaped pattern with a clear gap between the non-replicating G1 phase and G2 phase population at the feet of the arc, which demonstrates efficient Cy3-xx-dUTP incorporation in nearly all S-phase cells (Fig. 1C). Comparison of Cy3-xx-dUTP with a related

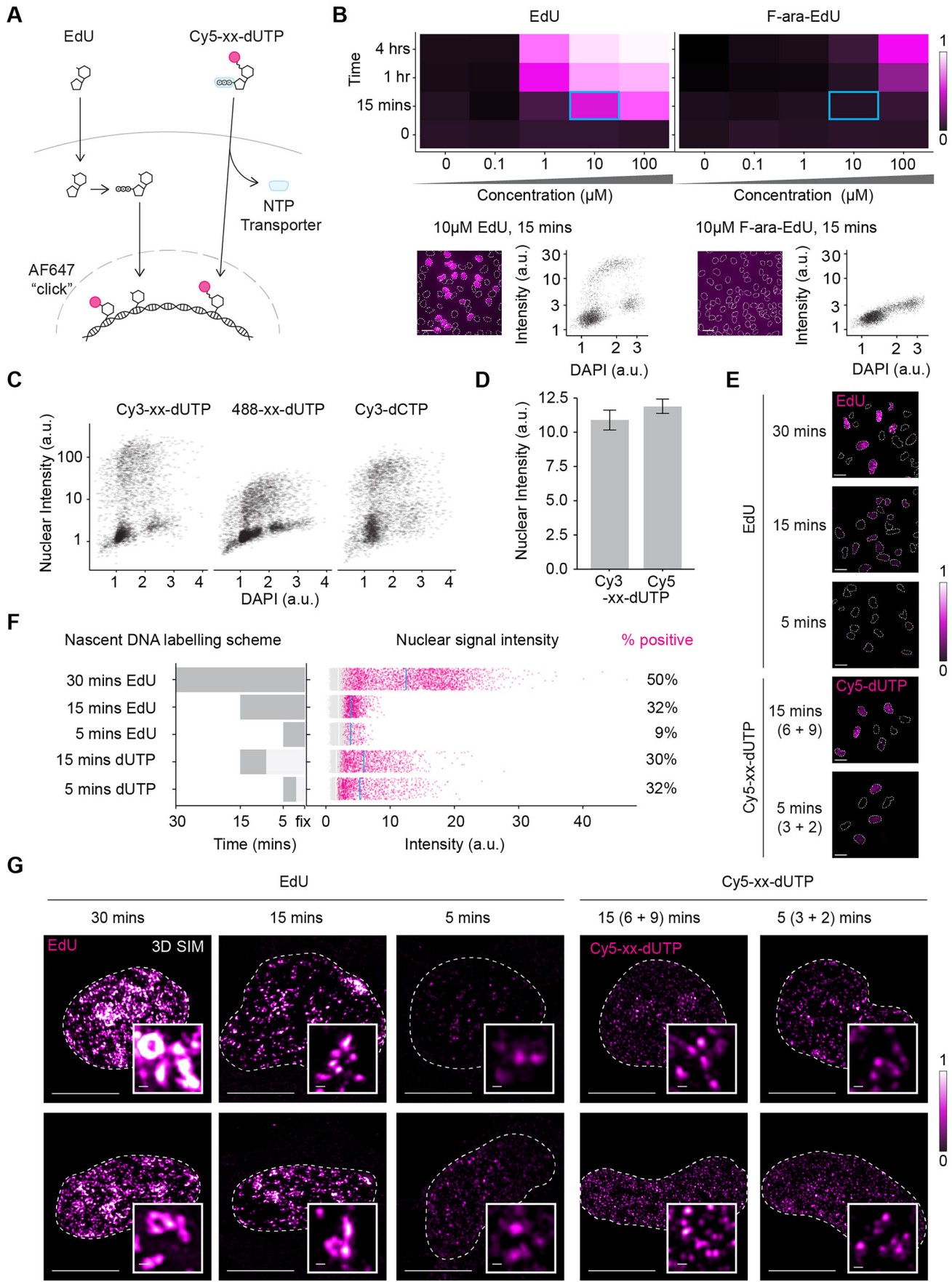

**Figure 1. Optimised nascent DNA labelling for nanoscale imaging.**

(A) Schematic diagram of two distinct approaches for fluorescent labelling of nascent DNA showing the transport of EdU and fluorescently linked dUTPs (e.g. Cy5-xx-dUTP) into the nucleus. EdU diffuses across the membrane into the cytoplasm, where it is tri-phosphorylated before becoming available for incorporation into replicating DNA. Following EdU incorporation and cellular fixation, a fluorescent azide-dye (e.g. AF647) is clicked to EdU for detection. Cy5-xx-dUTPs are transported into the cell via a nucleoside triphosphate (NTP) transporter, where they are immediately available for incorporation into replicating DNA and detection. Fluorescent dyes are depicted by the magenta circles. (B) Heatmaps depicting a time-concentration matrix for the fluorescent nuclear signal intensities upon EdU and F-ara-EdU treatment. Each tile is colour-scaled according to the average incorporated fluorescent signal, calculated as the 95th percentile minus the 5th percentile of three independent experiments. Below are representative images (nuclei are annotated with a white dashed line, scale bar: 20 μm) and representative scatterplots of nuclear integrated intensities (arbitrary units [a.u.]) against DAPI nuclear integrated intensities (a.u.) of individual cells treated with 10 μM EdU and F-ara-EdU for 15 min. These conditions are annotated by a cyan rectangle on the heatmap. (C) Scatter plots of Cy3-xx-dUTP, 488-dUTP and Cy3-dCTP nuclear integrated intensities (a.u.) against DAPI nuclear integrated intensities (a.u.) of individual cells. Cells were labelled for 60 (4 + 56)-min with 6 μM fluorescent nucleotides. Data from two independent experiments are pooled, $n = $ >3500 cells. (D) Bar plot displaying the mean nuclear integrated intensities (a.u.) for Cy3-xx-dUTP and Cy5-xx-dUTP from three independent experiments, error bars represent SEM. Cells were labelled for 15 (4 + 11)-min with 6 μM fluorescent nucleotides. (E) Representative confocal microscopy images after treatments as described in 1F. Nuclei are annotated with a white dashed line, scale bars: 20 μm. (F) Left bar plot: nascent DNA labelling scheme for different EdU (10 μM) and dUTP (3 μM) pulse durations. Dark grey bars indicate the pulse duration, and light grey bars indicate the chase duration. Right bar plot: jitter plots illustrate nuclear integrated signal intensities (a.u.) after treatments with the corresponding DNA labelling scheme. Negative cells in grey (mean nuclear integrated intensity shown as a white line) and positive S-phase cells in magenta (mean nuclear integrated intensity shown as a cyan line). Normalised data were pooled from two independent experiments, and a fixed threshold function was used to stratify cells into either negative cells or positive S-phase cells. For each condition, the percentage of signal-positive cells is indicated on the right. (G) Representative 2D maximum projections of 3D-SIM images after treatments as described in 1F. Nuclei are annotated with a white dashed line, scale bars: 10 μm. Insets display representative events, scale bars: 250 nm. Source data are available online for this figure.

infrared variant, Cy5-xx-dUTP, confirmed these results and demonstrated similar labelling proficiencies (Fig. 1D). Since the latter two nucleotide variants have a long, flexible aminoallyl linker (indicated by xx) between the fluorescent dye and dUTP, we wondered if exclusion of this linker would change DNA labelling efficacy. We found Cy3-xx-dUTP and Cy5-xx-dUTP to yield a small but significant increase in nuclear signal intensities compared to their non-linker counterparts (Fig. EV1B,C). Based on these findings, we selected fluorescent dUTPs with a linker for subsequent experiments.

We then investigated how varying pulse durations affected nuclear signal intensity and the detection of replication nanostructures. Increasing the EdU pulse from 5 to 30 min resulted in a significant rise in signal, with the 30-min EdU pulse achieving a 4-fold higher median nuclear intensity and a 6-fold increase in the percentage of foci-positive cells compared to the 5-min EdU pulse (Fig. 1E,F). Notably, using the same detection wavelength and imaging conditions, a 5 (3 + 2)-min Cy5-xx-dUTP treatment yielded 32% foci-positive cells, while a 5-min EdU pulse only yielded 9% foci-positive cells. This observation implies that the metabolic conversion of EdU into an active nucleotide triphosphate, a step critical for its incorporation into genomic DNA, becomes limiting when pulse periods are short. A time-series QIBC experiment using incremental EdU pulse lengths (5–120 min) confirmed this notion and indicated that the relationship between EdU pulse duration and EdU integrated intensity becomes non-linear for pulses shorter than 1 h (Fig. EV1D) (Pereira et al, 2017). Extending the Cy5-xx-dUTP pulse from 5 (3 + 2) min to 15 (6 + 9) min increases the median signal intensity by 15%. Irrespective of the pulse duration, the SNTT1-based delivery method resulted in a wider range of nuclear intensities compared to the EdU-based method (Fig. 1F), likely reflecting intercellular variation in transfection efficiency. Nonetheless, a 15-min pulse of Cy5-xx-dUTP was sufficient to generate a significant fraction (30%) of foci-positive cells reaching nuclear intensities never observed upon a 15-min EdU pulse (Fig. 1F). 3D-SIM imaging of the resulting labelled nascent DNA structures revealed a progression in morphology that corresponds to pulse length: short pulses yielded small, dot-like

foci, while longer pulses resulted in larger, loop-like structures (Fig. 1G). While dot-like signals are well-suited for quantitative measurements with high accuracy, the loop-like signals provide added structural context. These studies demonstrated different pros and cons of the nascent labelling strategies, with SNTT1-based methods delivering the highest signals for short pulse periods and EdU-based protocols excelling in intercellular consistency, ease-of-use, and dye-flexibility. Together, these findings provide a basic framework for nascent DNA labelling strategies in human cells and reveal practical parameters to detect and resolve nanoscale replication events.

## Dual-colour nascent DNA labelling for 3D-SPARK

Since the 15-min Cy5-xx-dUTP transfection efficiently labelled S-phase cells and provided well-defined replication foci, we sought to quantify the extent of DNA labelling during such pulse conditions. To internally validate the labelled events, cells were pulse-labelled with a 1:1 mixture of Cy3-xx-dUTP and Cy5-xx-dUTP for a total of 15 (4 + 11)-min and analysed in parallel by 2D DNA fibre assays and 3D-SIM. The labelled tracks on DNA fibres had a mean length of approximately 6 nm, which corresponds to ~16 kb of nascent DNA based on the reported fibre extension rate of 2.59 kbp/μm (Jackson and Pombo, 1998) (Fig. 2A). Considering the mean inter-origin distance is ~100 kb in RPE1 cells, such fibre tract dimensions could support the resolution of individual replicons (Böhly et al, 2022; Koundrioukoff et al, 2023). Using 3D-SIM to visualise the same labelling indeed showed abundant dot-like foci in early S-phase nuclei, which, as expected, showed a high degree of overlap between Cy3 and Cy5 signals (Fig. 2B). Subsequent pulse-chase experiments revealed that this signal overlap was diminished when a 30-min chase period was introduced between the Cy3-xx-dUTP and Cy5-xx-dUTP pulses (Fig. EV2A). To confirm if our approach can detect dynamic DNA replication events compatible with click chemistry, we modified the setup to include sequential pulse labels using the NTP transporter and two distinct nucleotide triphosphates. To optimise replication fidelity, we started with EdUTP (the triphosphate form of EdU), as

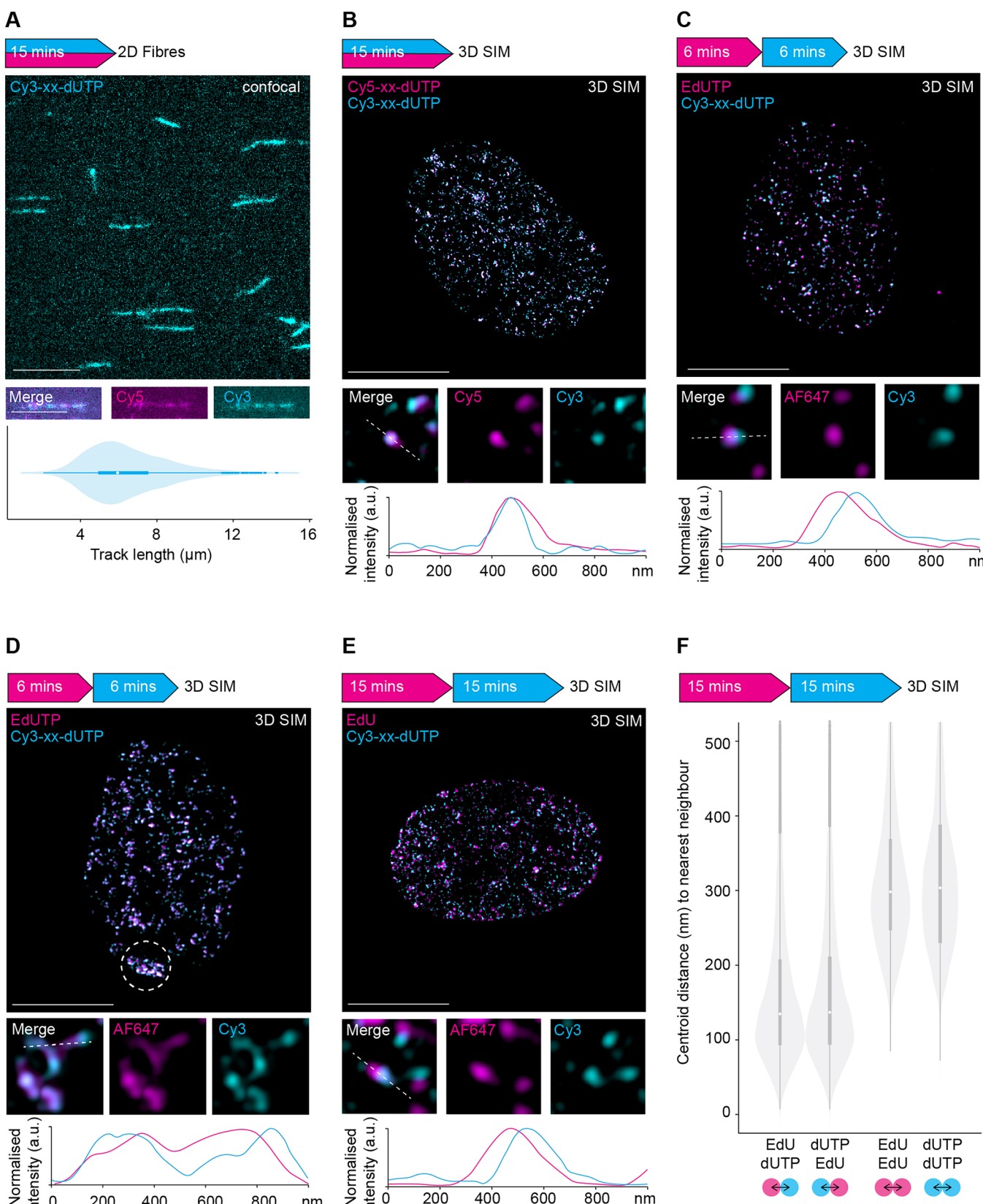

◀

**Figure 2. Dual-colour nascent DNA labelling for 3D-SPARK.**

The arrow depicts the overall experiment setup, including pulse timings, order of nascent DNA labels and detection method. (A) Representative image of fluorescent DNA fibres after a simultaneous 15 (4 + 11)-minute pulse using Cy5-xx-dUTP and Cy3-xx-dUTP at a 1:1 ratio, scale bar = 10 μm. An example of a dual colour nascent DNA tract is shown below, scale bars = 5 μm. The violin/box plot shows the DNA fibre tract length distribution from >450 Cy3-positive forks. Data distributions are shown as combined violin and box plots. Violin plots depict the distribution of DNA fibre tract length. The box shows the interquartile range (IQR) from the 25th (Q1) to the 75th (Q3) percentile; the white dot in the box corresponds to the median. Whiskers extend to the minimum and maximum values within 1.5 × IQR from Q1 and Q3, and grey dots outside this range represent potential outliers. (B) Representative 2D maximum projection of a 3D-SIM image of an early S-phase nucleus after a simultaneous 15 (4 + 11)-min pulse using Cy5-xx-dUTP and Cy3-xx-dUTP at a 1:1 ratio, scale bar = 10 μm. An example of a dual colour replication event is shown below. The line graph shows the normalised signal intensity profile across the depicted dual colour event, illustrated by the white dashed line in the merged image. (C) Representative 2D maximum projection of a 3D-SIM image of an early S-phase nucleus after a 6 (3 + 3)-minute EdUTP pulse followed by a 6 (3 + 3)-min Cy3-xx-dUTP pulse, scale bar = 10 μm. An example of a dual colour replication event is shown below. The line graph shows the normalised signal intensity profile across the depicted dual colour event, illustrated by the white dashed line in the merged image. (D) Representative 2D maximum projection of a 3D-SIM image of a late S-phase nucleus after a 6 (3 + 3)-min EdUTP pulse followed by a 6 (3 + 3)-min Cy3-xx-dUTP pulse, scale bar = 10 μm. An example of a dual colour replication event is shown below. The line graph shows the normalised signal intensity profile across the depicted dual colour event, illustrated by the white dashed line in the merged image. The white dashed circle indicates foci dense regions, likely reflecting synchronous DNA synthesis on the inactive X chromosome in mid/ late S-phase. (E) Representative 2D maximum projection of a 3D-SIM image of an early/ mid S-phase nucleus after a 15-min EdU pulse followed by a 15 (4 + 11)-min Cy3-xx-dUTP pulse (standard pulsing scheme), scale bar = 10 μm. An example of a dual colour replication event is shown below. The line graph shows the normalised signal intensity profile across the depicted dual colour event, illustrated by the white dashed line in the merged image. To complement the 2D maximum projection, the individual Z-planes are visualised in Movie EV1. (F) Data distributions are shown as combined violin and box plots. Violin plots show distances between centroids from each EdU focus to the closest dUTP focus, and vice versa, as well as between proximal EdU foci pairs and between proximal dUTP foci pairs. The box shows the interquartile range (IQR) from the 25th (Q1) to the 75th (Q3) percentile; the white dot in the box corresponds to the median. Whiskers extend to the minimum and maximum values within 1.5 × IQR from Q1 and Q3, and dots outside this range represent potential outliers. Source data are available online for this figure.

it contains a relatively small ethynyl group and allows efficient CuAAC click-labelling post fixation. We transfected RPE1 cells first with EdUTP for 6 (3 + 3)-min and then with Cy3-xx-dUTP for 6 (3 + 3)-min, followed by cell fixation, AF647-azide click-labelling and 3D-SIM imaging. The resulting nuclei showed well-defined replication foci in early S phase and the general nuclear foci patterns of the two nascent DNA labels matched per cell (Fig. 2C). At the nanoscale, the AF647 and Cy3 signals partially overlapped or often flanked each other, indicative of continued DNA synthesis during the dual pulse period (Fig. 2C). We observed similar dual colour events in mid/late S-phase nuclei, but, as expected (Casas-Delucchi et al, 2011; Chagin et al, 2016) these patterns were accompanied by prominent clustering of a substantial portion of nanofoci in concatenated loop like-structures and dense chromosome territories (Fig. 2D). Since event clustering compromises event separation and accurate replication kinetics inference, we focused our further 3D analysis on nuclei showing early S-phase patterns. Nevertheless, these observations, demonstrate that multiple NTP transfections are feasible and allow detection of multicolour DNA synthesis events in human cells.

We anticipated that replacing the first EdUTP transfection by direct EdU addition could further improve assay adaptability and consistency among single cells. Reducing transfection steps and media exchanges will ease applications that involve time-sensitive drug treatments and will further minimise variability related to transfection efficiency or cellular stresses. We thus performed sequential pulse labelling based on a short 15-min EdU pulse followed by a Cy3-xx-dUTP 15 (4 + 11)-min transfection. Subsequent AF647-azide click-labelling and 3D-SIM imaging revealed distinct dual-colour nanostructures reminiscent of the ones seen using the double-transfection setup (Fig. 2C,E; Movie EV1). Quantitative analysis of 17,000 segmented events per fluorescent label revealed an average nearest neighbour distance between AF647 and Cy3 foci of 135 nm, while events of the same type were more dispersed and showed an average nearest neighbour distance of 300 nm (Fig. 2F). Reciprocal nearest neighbour analysis of AF647 and Cy3 foci revealed a highly symmetrical relationship, indicating

that the two nascent DNA labels occur at relatively equal densities and show similar and consistent proximity patterns in the nucleus (Fig. 2F). These observations were further validated by pulse-chase experiments showing that a 30-min pulse between EdU and Cy3-xx-dUTP significantly increased the centroid distances between the two labels and diminished the signal overlap (Fig. EV2B). Given the suitability of the obtained object distances for quantitative 3D-SIM and the relative ease of sample preparation for yielding hundreds of multicolour replication nanostructures per cell, we selected the 15-min EdU followed by a 15 (4 + 11)-min Cy3-xx-dUTP treatment as the standard pulsing scheme in the 3D-SPARK protocol for further examination.

## Identification and quantitative analysis of basic replication structures

To identify nanoscale events suitable for replication dynamics inference, we developed an automated intensity-based analysis pipeline to identify, segment and classify dual-colour nanostructures containing EdU and dUTP signals. For replication dynamics inference, we focused our 3D analysis on early S-phase cells and high-confidence events, which excluded single-colour events, events of limited size (<50 voxels) or crowded and concatenated events consisting of >3 nanofoci. Although some of the excluded nanofoci represent genuine DNA synthesis events, we imposed high thresholds to obtain well-curated datasets of dual-colour events. Using these strict cut-offs, we kept ~50% of detected nanofoci for downstream analysis (Fig. EV3A–D). Upon sequential DNA labelling using the *standard pulsing scheme*, we identify up to 1539 dual-colour events in individual early S-phase RPE1 cells (on average 822 dual events per nucleus). Based on the observed 3D nanofoci patterns, DNA fibre data, and published estimates of early replication initiation zones in diploid human cell lines (Liu et al, 2021), we hypothesised that each dual-colour event might contain a single replication fork or replicon.

To stratify different types of dual-colour events, we classified three basic nanostructures termed initiation events, ongoing events,

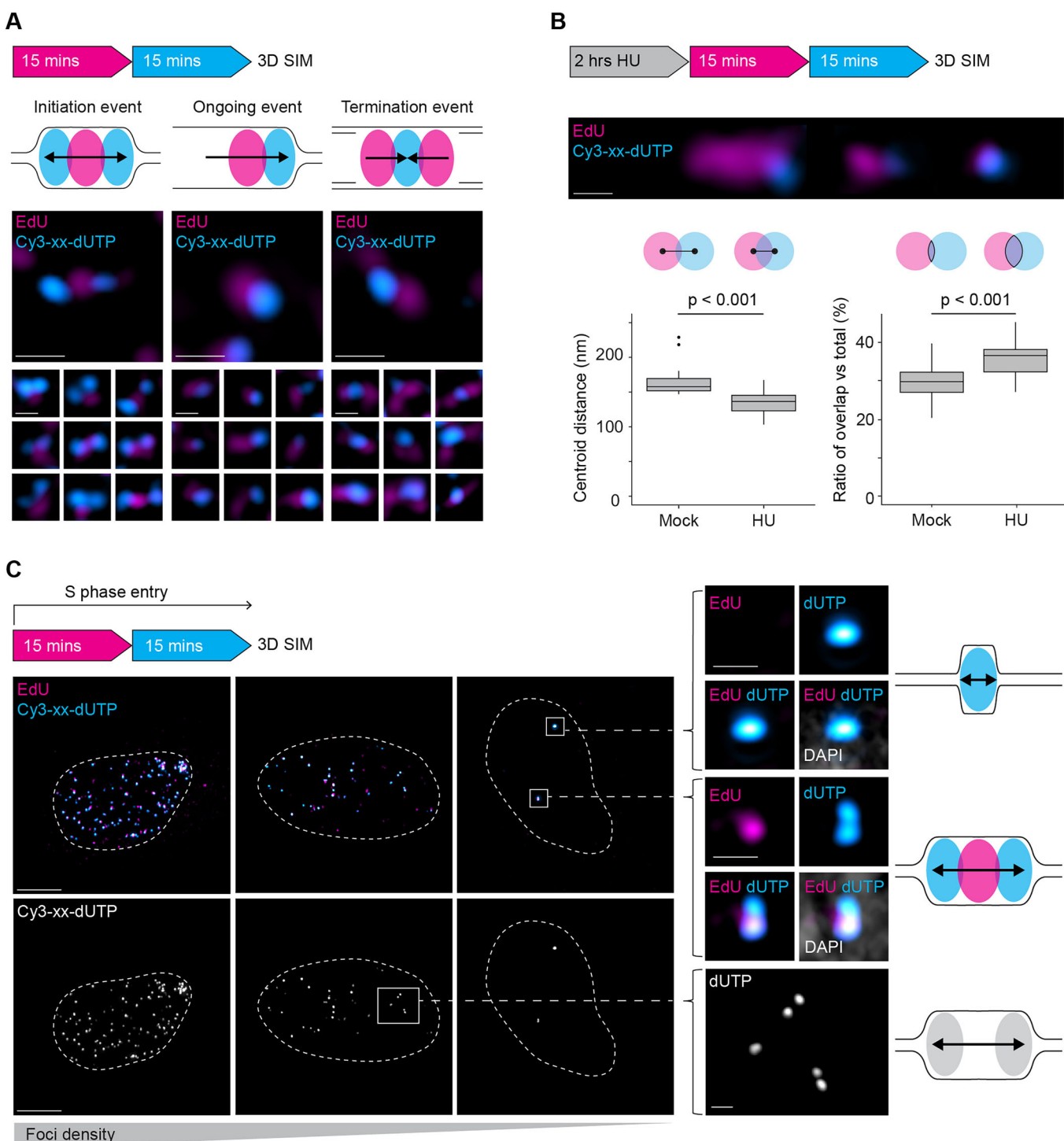

or termination events (Fig. 3A). *Ongoing events* are the most elementary structures, defined by one EdU focus associated with one dUTP focus. Initiation events have one EdU focus and two associated dUTP foci, possibly reflecting bi-directional DNA synthesis events fired within the dual-pulse window. Termination events have two EdU foci and one dUTP focus, indicative of the merging of two proximal DNA synthesis events. Under unchallenged conditions, the ratio of initiation, ongoing, and termination events in untransformed human RPE1 cells was 1:8:1, respectively (Fig. EV4A), which aligns with recent studies using automated events classifications on 2D DNA fibres (Li et al, 2022) and supports the notion that the majority of identified dual-colour events reflect single ongoing forks. Subsequent dye-swap experiments demonstrated similar event ratios and approximately 80% ongoing events (Fig. EV4B). To test if the classified *ongoing events* indeed represent ongoing DNA synthesis tracks, we blocked

**Figure 3.  Identification and quantitative analysis of basic DNA replication structures.**

(A) Illustrations of three basic replication structures: initiation, ongoing and termination events with representative 3D-SIM images, found following treatment with the *standard pulsing scheme*, scale bars: 250 nm. (B) Examples of long, medium and short dual-colour ongoing events imaged by 3D-SIM, scale bar: 250 nm. Box plots and illustrations showing the centroid distances between EdU and Cy3-xx-dUTP foci (bottom left) and the ratio of overlapping EdU and Cy3-xx-dUTP foci volume versus the total event volume (bottom right) in cells treated with or without HU and labelled with the standard pulsing scheme. Number of nuclei analysed: 28 (mock); 22 (HU). For each box, the inner line indicates the median, and the box limits show the 25th and 75th percentiles. Whiskers extend to edge values within 1.5 times the interquartile range between 25th and 75th percentiles from the box limits. Dots represent values beyond whisker range. *P* values were calculated with the Mann–Whitney *U*-test and adjusted for multiple testing using Bonferroni correction. $P = 4.5e\text{-}8$ (left graph), $P = 7.4e\text{-}5$(right graph). (C) Representative 2D maximum projections of 3D-SIM images of cells entering S-phase during treatment with the standard pulsing scheme. Upper panels show dual colour images of three nuclei with decreasing foci density. Lower greyscale panels show only the Cy3-xx-dUTP signal of the same nuclei, scale bars: 5 µm. Detailed images and illustrations on the right show individual replication events, with either a single Cy3-xx-dUTP or symmetric pair of Cy3-xx-dUTP foci, scale bars: 500 nm). Source data are available online for this figure.

replication fork progression by hydroxyurea (HU) and studied the nature of these nanostructures. HU treatment significantly reduced the centroid distance between pairs of EdU and dUTP foci and concomitantly increased the relative volume overlap of paired EdU and dUTP foci (Fig. 3B), demonstrating that the classified ongoing events represent dynamic DNA synthesis events and harbour multiple kinetics parameters at the nanoscale.

Given the sheer abundance of replication foci in human S-phase nuclei, we assessed whether tripartite structures, such as initiation events, are a biologically relevant distinct class of events or a mere result of foci crowding. To discriminate between these hypotheses, we labelled RPE1 cells using the standard pulsing scheme and screened for early S-phase cells with few replication foci (Fig. 3C). Despite the low density of replication foci, initiation events were readily detected and dUTP foci often appeared in pairs of similar intensity (Fig. 3C), arguing that initiation events represent elemental DNA replication structures featuring bi-directional DNA synthesis.

## Validation of dual-colour replication nanostructures by expansion microscopy

To independently validate the multicolour events observed by 3D-SIM, we developed an eXpansion Microscopy-based assay for replication kinetics (XMARK). Expansion microscopy enables the detection of nuclear nanostructures with minimal distortion by isotropically enlarging human cells using swellable hydrogels (Sun et al, 2021; Faulkner et al, 2022). Using established 3D tissue expansion protocols (Zwettler, Spindler, et al, 2020), we achieved uniform, four-fold nuclear expansion in human RPE1 cells (Fig. EV5A–D). Combining these protocols with EdU click-chemistry, we found that nuclear expansion increased the optical resolution while preserving the 3D spatial organisation of replication foci patterns (Fig. 4A). To establish dual-colour XMARK, we substituted Cy3-xx-dUTP from the standard pulsing scheme with dUTP-xx-STAR RED. The stable Abberior STAR RED dyes are more suitable for expansion microscopy than cyanine dyes (e.g. Cy3, Cy5 or AF647), which are prone to degradation during gel polymerisation (Wen et al, 2022; Kong et al, 2024). By performing XMARK on S-phase nuclei with low foci density, we observed common replication nanostructures that closely resemble those identified using 3D-SIM, including ongoing events, initiation events (Fig. 4B) and termination events (Fig. EV5E). These observations highlight the adaptability of our 3D-SPARK approach and substantiate the detection of elemental nascent DNA structures in human cells by two independent nanoscopy assays.

## 3D-SPARK is compatible with IF and direct correlation between DNA replication and damage biomarkers

To further characterise the dual-colour nascent DNA nanostructures, we combined our standard pulsing scheme with established immunofluorescence (IF) protocols to visualise replication fork proteins such as proliferating cell nuclear antigen (PCNA). At the single-cell level, EdU, Cy3-xx-dUTP and PCNA show highly similar foci patterns, affirming their specificity and shared localisation to active replication domains (Fig. 5A). In line with PCNA localising near the fork (and thus closest to the latest incorporated DNA), colocalisation analysis of 2D projected nuclei showed a higher correlation between PCNA and Cy3-xx-dUTP signals (mean Pearson's $R = 0.82$) than between PCNA and EdU signals (mean Pearson's $R = 0.61$) (Fig. 5B). Also at the nanoscale, the signals appeared adjacent rather than overlapping (Fig. 5C), as is expected for sequential labelling schemes and the transient binding kinetics of PCNA at nascent chromatin (Sirbu et al, 2012). The relative position of PCNA, Cy3-xx-dUTP, and EdU of individual events supported the spatial coherence and directionality of the replication nanostructures (Fig. 5C). The compatibility of 3D-SPARK with IF also allowed us to assess whether the dual-pulse labelling protocol induces DNA replication stress. To provide time for possible DNA damage signals to accumulate, we performed a 30-min EdU followed by a 30 (4 + 26)-min Cy3-xx-dUTP dual pulse treatment—twofold longer than the standard labelling scheme—and stained for two established DNA replication stress markers, phospho-Histone H2A.X (pH2AX) and phospho-RPA (pRPA). Despite the relatively long exposures, we did not detect a significant induction of pH2AX or pRPA signals in dual-pulsed S-phase cells compared to EdU-only controls, which was confirmed at the population level (Fig. 5D) and at the nanoscale (Fig. 5E, F). In contrast, when samples were pretreated with HU, a marked induction of pH2AX and pRPA signals were observed, which, as expected, correlated spatially with EdU and Cy3-xx-dUTP-labelled DNA (Fig. 5D–F). These findings indicate that labelling protocols, with treatment durations twice those of the standard pulsing scheme, do not trigger overt DNA damage or fork stalling, supporting the use of 3D-SPARK to study replication dynamics with high precision and at multiple scales in human cells.

## 3D-SPARK detects treatment-induced and local perturbations in replication initiation

Having validated the key elements of 3D-SPARK, we sought to investigate whether internal and external replication challenges

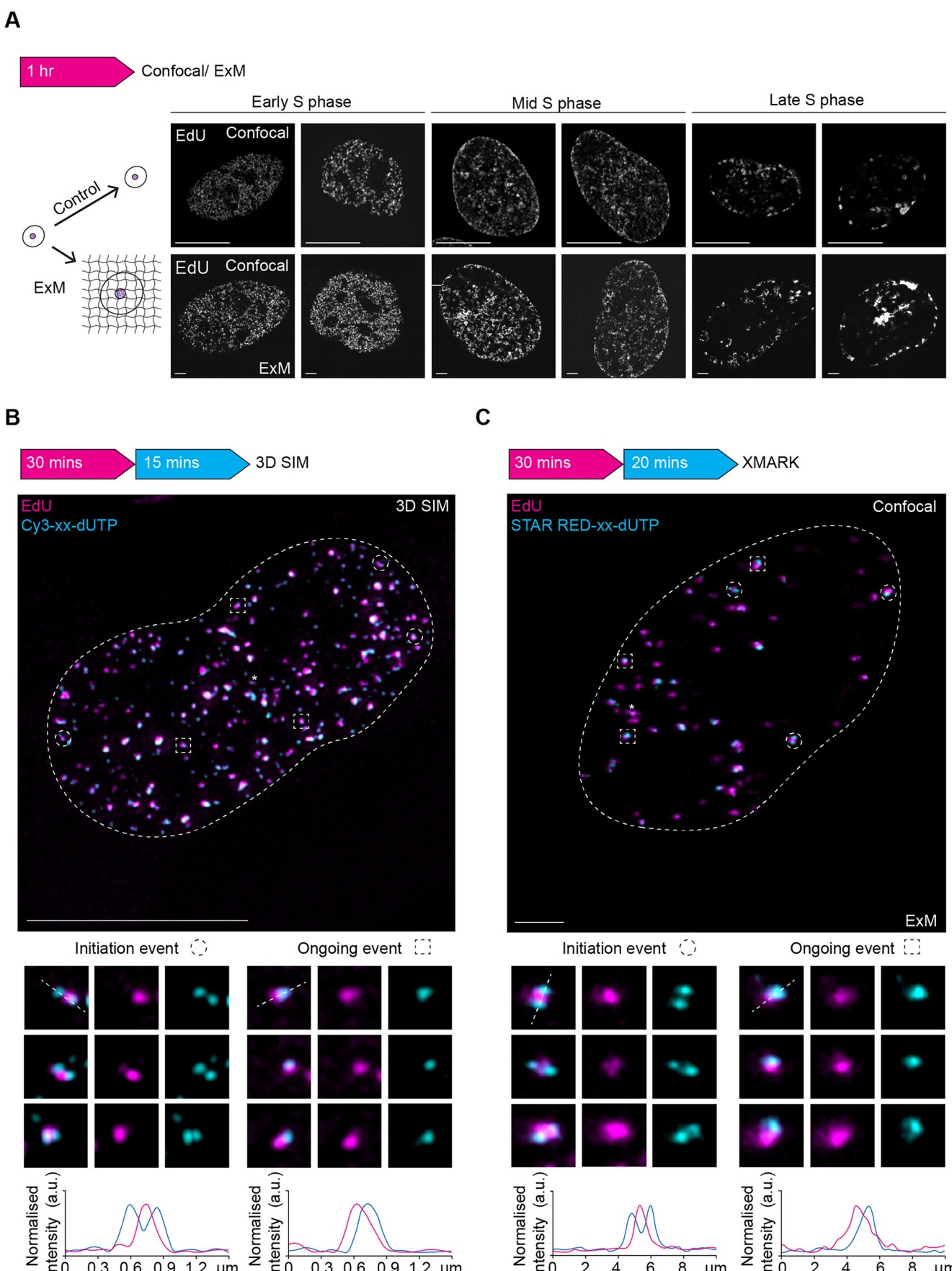

**Figure 4.  Validation of dual-colour DNA nanostructures by expansion microscopy.**

(A) Following a 1-h EdU pulse, fixed cells were either imaged using confocal microscopy or underwent cellular expansion before imaging. Overall foci patterns were maintained in expanded early, mid, and late S-phase cells, scale bars: 10 μm. (B) Representative 2D maximum projection of a 3D-SIM image of an early S-phase nucleus after a 30-min EdU pulse followed by a 15 (4 + 11)-min Cy3-xx-dUTP pulse, scale bar = 10 μm. The nuclear perimeter is annotated with a white dashed line and selected examples of initiation events are indicated by dashed circles, and ongoing events are indicated by dashed squares. For each example, an EdU only, Cy3-xx-dUTP only and merged channel image are shown below. The line graphs show the normalised signal intensity profile (a.u.) across the depicted dual colour events, illustrated by the white dashed line in the merged image. The asterisk indicates a termination event. (C) Representative 2D maximum projection of an XMARK image of an early S-phase nucleus after a 30-min EdU pulse followed by a 20 (4 + 16)-min STAR RED-xx-dUTP pulse, scale bar = 10 μm. The nuclear perimeter is annotated with a white dashed line, and selected examples of initiation events are indicated by dashed circles, and ongoing events are indicated by dashed squares. For each example, an EdU only, STAR RED-xx-dUTP only and merged channel image are shown below. The line graphs show the normalised signal intensity profile (a.u.) across the depicted dual colour events, illustrated by the white dashed line in the merged image. The asterisk indicates a termination event. Source data are available online for this figure.

shift the relative abundance of basic replication nanostructures in human cells. The efficacy of DNA synthesis can be influenced by numerous factors, including immediate stresses such as transcription-replication conflicts, oncogene expression or medication (Saxena and Zou, 2022). The chemotherapeutic agent HU is a potent inhibitor of fork progression (Fig. 3B) (Ge et al, 2007), which is predicted to increase initiation events—due to dormant origin firing and slow fork divergence—and reduce termination events—as forks struggle to reach each other (Fig. 6A). To directly test these predictions, we exposed RPE1 cells to 2 h of HU treatment followed by the standard pulsing scheme. Quantitative analysis of >19,000 elemental replication nanostructures revealed a significant shift towards initiation events at the expense of termination events (Fig. 6B).

To examine whether 3D-SPARK could also detect oncogene-induced changes in DNA replication dynamics, we used an established human bronchial epithelial cell model overexpressing oncogenic CDC6 at will (HBEC CDC6-tetON). Previous studies found CDC6 overexpression to cause a significant drop in EdU incorporation within 12 h and before the onset of DNA breaks or senescence (Zampetidis et al, 2021), which is in line with a model where deregulated CDC6 causes over-licensing and sequestration of limiting origin-firing factors (Seoane and Morgan, 2017) (Fig. 6C). QIBC analysis revealed that CDC6 overexpression caused a significant reduction in EdU intensity while elevating the levels of chromatin-associated MCM2 and PCNA, confirming that persistent CDC6 expression impairs DNA synthesis (Fig. EV6). We performed 3D-SPARK in HBEC CDC6-tetON cells and identified a significant drop in initiation events compared to ongoing events upon 24 h of CDC6 induction (Fig. 6D). In contrast to HU, CDC6 overexpression did only marginally affect the relative percentage of termination events (Fig. 6D). These results are consistent with a selective role of human CDC6 in controlling origin activity and demonstrate the feasibility of 3D-SPARK to visualise disease-related phenomena in different human cell models. Early replicating domains are often associated with open chromatin and active transcription, yet to avoid transcription-replication conflicts and R-loop-mediated DNA damage, replication initiation tends to be adjacent to but not directly overlapping with highly transcribed regions (Demczuk et al, 2012; Liu et al, 2024; Mesner et al, 2013; Petryk et al, 2016). To study the interplay between transcription and DNA replication, we combined 3D-SPARK with IF and simultaneously detected dual-coloured DNA synthesis events and SON proteins in RPE1 cells (Fig. 6E,F). SON proteins are biomarkers of membraneless RNA processing bodies known as nuclear speckles, which are proposed to locally enhance

transcription and DNA replication activities (Kim et al, 2020; Ilık et al, 2020; Gholamalamdari et al, 2025). Plotting the distances of the three different types of dual colour events relative to SON, suggested that DNA replication tends to initiate nearby SON and terminate further away (Fig. 6E), in line with nuclear speckles functioning as local stimuli for replication initiation. While DNA replication foci were readily detected in the vicinity of SON-speckles, direct comparisons of signal overlap between Cy3-xx-dUTP and SON foci, and Cy3-xx-dUTP and EdU foci, revealed significant exclusion of replication events within nuclear speckles, in line with these bodies consisting mainly of protein and RNA condensates and the avoidance of R-loop-mediated DNA damage in replicating cells (Fig. 6G,H). Together, these observations demonstrate the ability of 3D-SPARK to detect DNA synthesis events relative to IF biomarkers and highlight the capacity of human cells to adapt DNA replication dynamics to different cellular and pharmacological contexts.

## Replication foci patterns and drug responses at defined S-phase timings

Distinct patterns of replication foci within the nucleus serve as reliable indicators for predicting the timing of S-phase progression in unperturbed cells. These different spatial distributions of replication events create unique cellular liabilities, suggesting that early, mid, and late S-phase cells may respond differently to drug treatments. To study replication dynamics in a time-resolved manner, we combined 3D-SPARK with cell synchronisation using the CDK4/6 inhibitor Palbociclib. Short-term (1-day) Palbociclib treatment induces a reversible G1 arrest and has been shown to generate highly synchronised RPE1 cell cultures (Trotter and Hagan, 2020). To study how early-S phase and mid/late S phase cells respond to DNA replication stress, we fixed RPE1 cells 4 or 8 h post Palbociclib release and exposed the cells to HU 2 h prior to fixation (Fig. 7A). Subsequent 3D-SIM analysis confirmed efficient release and progression through S phase as the nuclei showed abundant DNA synthesis and the characteristic replication foci patterns associated with early S phase and mid/late S phase (Fig. 7B). Automated foci quantification and event segmentation revealed a significant increase in EdU and Cy3-xx-dUTP foci within crowded events 8 h post Palbociclib release compared to 4 h post Palbociclib release (Fig. 7C), in line with spatial clustering of late-replicating regions (Casas-Delucchi et al, 2011; Chagin et al, 2016). While this result confirmed data crowding as an inherent challenge of studying replication dynamics in late S-phase cells, it also indicated that 3D SPARK remained able to identify >200

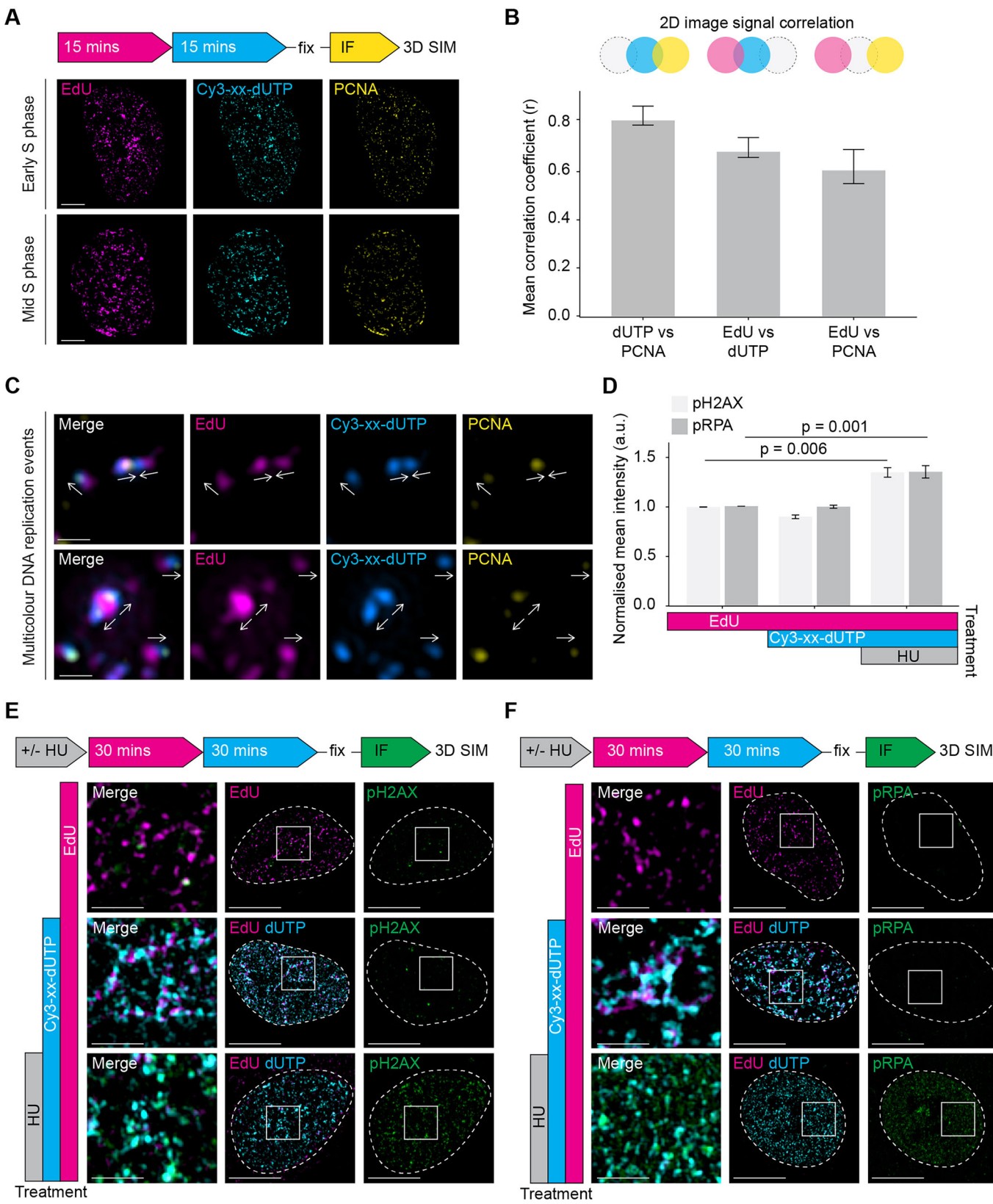

◄  **Figure 5.  3D-SPARK is compatible with IF and direct correlation between DNA replication and damage biomarkers.**

(A) Representative 2D maximum projections of 3D-SIM images of an early and mid S-phase nucleus after labelling with the standard pulsing scheme and post-fixation PCNA immunofluorescence staining, scale bar = 5 µm. (B) Bar plot showing the mean correlation coefficient (Pearson's *R* value with no threshold) between indicated image channels, as determined by the FIJI plugin Coloc 2, *n* = 4. Error bars represent SD. (C) Examples of nuclear areas containing individual replication events, with arrows indicating inferred directionality, scale bar = 500 nm. For each area, an EdU only, Cy3-xx-dUTP only, PCNA only and merged channel image are shown below. (D) Bar plot showing the mean nuclear integrated intensities (a.u.) of pH2AX and pRPA for cells treated as described in 1E, F, respectively. Three replicates per condition, error bars represent SD. (E) Representative 2D maximum projections of 3D-SIM images of cells after three different treatments: EdU for 30-min only, EdU for 30-min and Cy3-xx-dUTP for 30 (4 + 26)-min or 2-h HU treatment prior to labelling with EdU for 30-min followed by Cy3-xx-dUTP for 30 (4 + 26)-min. All conditions underwent post-fixation immunofluorescence staining of pH2AX, scale bars: 10 µm. The nuclear perimeter is annotated with a white dashed line, and a selected area (white square) is shown in the merged image, scale bars: 2 µm. (F) Representative 2D maximum projections of 3D-SIM images of cells after three different treatments: EdU for 30-min only, EdU for 30-min and Cy3-xx-dUTP for 30 (4 + 26)-min or 2-h HU treatment prior to labelling with EdU for 30-min followed by Cy3-xx-dUTP for 30 (4 + 26)-min. All conditions underwent post-fixation immunofluorescence staining of pRPA, scale bars: 10 µm. The nuclear perimeter is annotated with a white dashed line, and a selected area (white square) is shown in the merged image, scale bars: 2 µm. Source data are available online for this figure.

individual dual-colour events per cell, allowing quantitative nanoscale DNA synthesis measurements in a time-resolved manner. Notably, we found HU to trigger a significant increase in initiation events, especially in 8 h post Palbociclib release, while significantly reducing the frequency of termination events irrespective of S phase timing (Figs. 7D and EV7). These observations are in line with our previous results in non-synchronised cells (Fig. 6A) and support a model where fork stalling prevents DNA synthesis events to converge and triggers dormant origin firing through Cyclin-A-CDK activities that progressively increase in S-phase (Lemmens et al, 2018, Katsuno et al, 2009).

### RIF1 loss changes DNA replication dynamics within entire nuclear subdomains as well as the 3D constellation of individual replication nanostructures

The human genome undergoes major rearrangements in 3D architecture during development and disease, imposing a fundamental challenge to DNA replication. To see if 3D-SPARK could provide quantitative insight into these spatial adaptations, we focused on the role of RIF1, a well-conserved regulator of 3D genome organisation and DNA replication timing. We noticed that RIF1 deficiency (either by transient depletion or mutation) caused an apparent lack of mid-S phase foci patterns that are characterised by the enrichment of replication events close to nucleoli and the nuclear lamina (Fig. 8A,B). Quantification of the radial distribution of nuclear replication events in human RPE1 cells confirmed a selective loss of replication nanofoci near the nuclear periphery (Fig. 8C). Characterisation of the relative frequencies and spatial dimensions of dual-colour events revealed smaller yet more abundant replication structures in RIF1 deficient cells compared to RIF1 proficient controls (Fig. 8D,E). Within ongoing events the centroid distances between EdU and Cy3-xx-dUTP foci were slightly smaller in RIF1 deficient cells, but the relative signal overlap was not increased (Fig. 8F,G). These data are consistent with RIF1 controlling replication timing mainly at the level of spatial organisation instead of directly controlling fork speed (Yamazaki et al, 2012; Cornacchia et al, 2012). While exploring the rich array of foci patterns in 2D maximum projections of both normal and RIF1 depleted conditions, we noticed a constellation of initiation events in RIF1 depleted cells that was rare in control cells, i.e. initiation events with an EdU focus at the centre and two Cy3-xx-dUTP foci positioned almost opposite each other (Fig. 8H). Overlaying initiation events in RIF1 depleted and control cells indeed suggested RIF1 restrained the angle between Cy3-xx-dUTP foci (Fig. 8H). Independently, automated analysis of 3D centroid

positions in RIF1 proficient and RIF1 mutant cells confirmed these results, revealing a significant widening of the angle between Cy3-xx-dUTP foci of individual initiation events (Fig. 8I). These conformational changes in replication nanostructures are in line with RIF1's role in confining 3D genome architecture and limiting chromatin loop sizes (Yamazaki et al, 2012), and highlight the ability of 3D-SPARK to reveal numerical as well as 3D structural changes in DNA replication.

## Discussion

The timing and dynamics of DNA replication correlate with chromosome position, topological domains and the proximity to the nuclear lamina, suggesting that DNA synthesis is controlled by spatial cues from 3D nuclear structures (Marchal et al, 2019; Chen and Buonomo, 2023). Current approaches based on sequencing, DNA fibres, or single-cell imaging, however, lack the spatial resolution to analyse individual replication events within the 3D cellular context (Vengrova and Dalgaard, 2009; Fajri and Petryk, 2024). Here we describe 3D-SPARK, a new assay based on nanoscopy of multi-colour nascent DNA synthesis tracts that allows direct visualisation and retrospective assessment of DNA replication dynamics in situ (Figs. 1, 2). This approach provides quantitative measurements of thousands of well-defined replication nanostructures per experiment and enables stratification of basic DNA replication events, allowing researchers to directly link replication dynamics to biomarkers in the same cell or condition (Figs. 3–7).

We established 3D-SPARK using EdU click chemistry and direct transfection of fluorescent dNTPs by SNTT1 (Figs. 1, 2). To facilitate its implementation, we built the assay using efficient and commercially available reagents that are compatible with standard immunofluorescence protocols and multiple cell types (Figs. 5–7). Alternative dNTP delivery strategies such as microinjection are highly precise, but it is technically demanding and not suitable for high-throughput experiments. Other strategies such as hypotonic shock, electroporation, scratch loading, bead loading or lipid-based transfection methods allow for simultaneous dNTP delivery to multiple cells but can induce significant cellular stress (Stewart et al, 2018). SNTT1-based dNTP delivery is suitable for high-throughput flow cytometry and microscopy studies and enables effective labelling of the entire S-phase population with minimal cytotoxicity in a variety of cancer cell lines and human primary fibroblasts (Kužmová et al, 2021). An illustrated, step-by-step 3D-SPARK protocol is provided as Appendix Protocol.

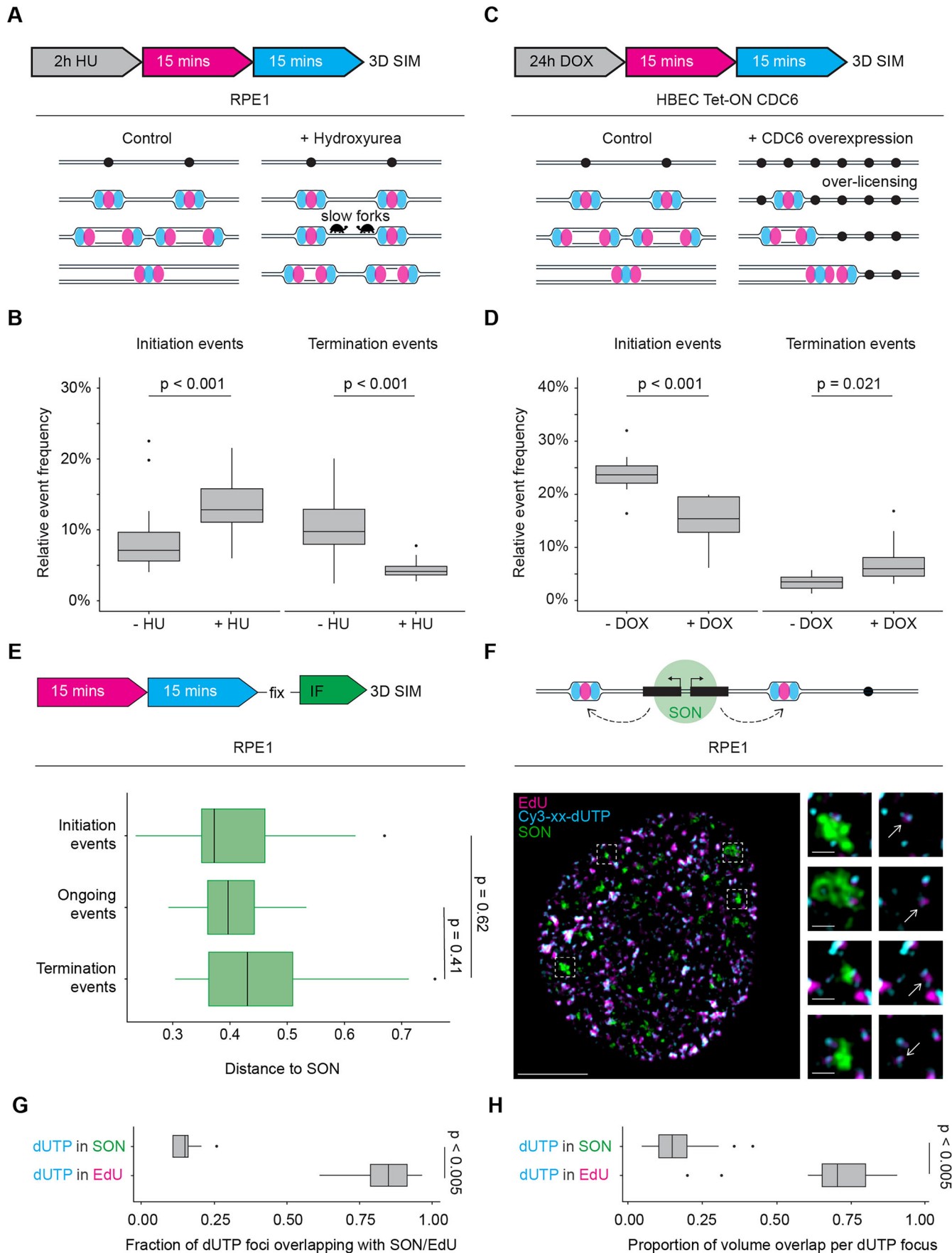

**Figure 6. Internal and external replication challenges shift nanoscale replication dynamics.**

(A) Treatment scheme and working model for hydroxyurea (HU) induced replication dynamics changes in RPE1 cells. In control conditions, replication origins (black circles) fire and are labelled with sequential EdU and Cy3-xx-dUTP pulses. These replication forks elongate until adjacent replication forks meet and terminate. When treated with HU, replication origins fire, yet forks proceed at a slower speed. (B) Box plot showing the relative event frequency of initiation and termination events of cells treated according to 6 A. Number of nuclei analysed: 26 (mock); 19 (HU). For each box, the inner line indicates the median, and the box limits show the 25th and 75th percentiles. Whiskers extend to edge values within 1.5 times the interquartile range between 25th and 75th percentiles from the box limits. Dots represent values beyond whisker range. $P$ values were calculated with the Mann–Whitney $U$-test and adjusted for multiple testing using Bonferroni correction; $p = 1.0e-4$ (initiation), $p = 1.7e-6$ (termination). (C) Treatment scheme and working model for replication dynamics alterations upon CDC6 oncogene induction by 24-h doxycycline (DOX) treatment in HBEC Tet-ON CDC6 cells. Control conditions are the same as 7A. DOX treatment results in CDC6-induced over-licensing and firing factor exhaustion, resulting in reduced initiation. (D) Box plot showing the relative event frequency of initiation and termination events of cells treated according to 6C. Number of nuclei analysed: 14 (no doxycycline); 15 (24 h doxycycline). For each box, the inner line indicates the median, and the box limits show the 25th and 75th percentiles. Whiskers extend to edge values within 1.5 times the interquartile range between 25th and 75th percentiles from the box limits. Dots represent values beyond whisker range. $P$ values were calculated with the Mann–Whitney $U$-test and adjusted for multiple testing using Bonferroni correction; $p = 2.9e-6$. (E) Box plot of the relative initiation event frequency, relative to SON speckles, following treatment with the standard pulsing scheme and post-fixation SON immunofluorescence staining. Number of nuclei analysed: 25. For each box, the inner line indicates the median and box limits show 25th and 75th percentiles. Whiskers extend to edge values within 1.5 times the interquartile range between 25th and 75th percentiles from the box limits. Dots represent values beyond whisker range. $P$ values were calculated with the Mann–Whitney $U$-test and adjusted for multiple testing using Bonferroni correction. (F) Representative 2D maximum projection of a 3D-SIM image of cells treated according to 6E, scale bar: 5 μm. The regions indicated by dashed squares are shown to the right, where arrows indicate examples of initiation events in close proximity to SON speckles, scale bars: 500 nm. Above, the SON model is illustrated, where SON speckles promote origin firing. (G) Box plot showing the fraction of Cy3-xx-dUTP foci overlapping with SON or EdU on the total volume of foci, $n = 25$ cells. For each box, the inner line indicates the median, and the box limits show the 25th and 75th percentiles. Whiskers extend to edge values within 1.5 times the interquartile range between 25th and 75th percentiles from the box limits. Dots represent values beyond whisker range. $P$ values were calculated with the Mann–Whitney $U$-test and adjusted for multiple testing using Bonferroni correction; $p = 1.6e-14$. (H) Box plot showing the proportion of overlapping volume per dUTP-xx-Cy3, focus on the total volume of the SON or EdU foci it overlaps with. Each value represents the median per nucleus analysed ($n = 25$). For each box, the inner line indicates the median, and the box limits show the 25th and 75th percentiles. Whiskers extend to edge values within 1.5 times the interquartile range between 25th and 75th percentiles from the box limits. Dots represent values beyond whisker range. $P$ values were calculated with the Mann–Whitney $U$-test and adjusted for multiple testing using Bonferroni correction; $p = 1.1e-12$. Source data are available online for this figure.

Several studies have exploited the relative ease, cost-effectiveness and flexibility of EdU click chemistry to study genome biology at the nanoscale. EdU can be detected directly using fluorescent dyes or indirectly by IF, making it compatible with a wide range of super-resolution imaging modalities (Chagin et al, 2016; Cseresnyes et al, 2009; Roy et al, 2018; Su et al, 2020; Whelan et al, 2020; Zessin et al, 2012). 3D-SPARK combines EdU and SNTT1-based DNA labelling and does not require transgenic reporters (e.g. tagged PCNA (Chagin et al, 2016), or fork stalling agents (Triemer, et al, 2018; Whelan et al, 2020), which makes it suitable for studying ongoing DNA synthesis in diverse settings. The adaptability of our multi-colour DNA labelling strategy is also highlighted by the fact that we employed 3D-SPARK using two fundamentally different nanoscale imaging modalities: 3D-SIM and expansion microscopy (Fig. 4). Expansion microscopy allows multi-colour imaging of nanostructures without the need of image reconstruction algorithms, which makes it highly complementary to 3D-SIM approaches. 3D-SIM allows researchers to use a wide palette of fluorophores, while the expansion-microscopy-based assay for replication kinetics (XMARK) can be done using conventional, diffraction-limited microscopes, further increasing the accessibility of 3D replication exploration.

3D-SPARK detects DNA synthesis events within the cell's native chromatin architecture, allowing direct quantification and comparisons of local replication signals. The tight relationship between DNA replication dynamics and 3D nuclear architecture is underlined by the dual role of RIF1, which controls both RT and 3D nuclear organisation (Fig. 7) (Gnan et al, 2021). Future efforts to combine 3D-SPARK with genomic localisation methods (e.g. Fluorescence In Situ Hybridisation) (Beckwith et al, 2025) and cellular activity sensors (Lemmens et al, 2018) could help in elucidating the many layers of RT regulation. Since the intensity and size of nascent DNA signals are subject to chromatin compaction, complementary single-cell or single-molecule

sequencing approaches can be helpful to provide replication kinetics information independent of chromatin status (Theulot et al, 2022; van den Berg et al, 2024). By combing drug treatments with our standard pulsing scheme we found HU to significantly reduce the mean intensities of EdU but not of dUTP foci, in line with HU being present throughout the entire EdU pulse and DNA synthesis rapidly restoring upon dUTP transfection when HU is removed (Fig. EV7). Building on the findings presented here and earlier work using 2D fibre assays (Liu, 2021), we envision that 3D-SPARK can be modified to address various aspects of DNA replication stress, including fork restart and degradation kinetics. In line with our observations (Fig. 6), recent multi-omics studies have identified striking correlations between DNA replication initiation and active transcription and found that PCNA foci in early-S phase cells preferentially occur near SON-speckles (Chen et al, 2019; Kim et al, 2020; Gholamalamdari et al, 2025). With transcription-replication conflicts recognised as drivers of cancer genomic instability, tumour evolution and therapy resistance, the interplay between local transcription activities and DNA replication kinetics remains an active field of research (Tang et al, 2024). While our 3D-SIM and QIBC data did not reveal significnant DNA damage or fork stalling using our standard pulsing scheme (Fig. 5), we noticed a modest change in dual-colour event ratios in our dye-swap studies (Fig. EV4B), possible hinting toward altered replication progresson or fork symmetry by Cy3-xx-dUTP. We thus recommend using the standard pulsing scheme for 3D-SPARK (Fig. 2E) and in general to use the least bulky nucleotide analogue first when performing replication dynamics studies.

The abundance of DNA synthesis sites in a replicating human cell remains a key challenge in the field, irrespective of whether one works with DNA sequencing, DNA fibres or imaging approaches, which is particularly challenging in late S-phase cells when DNA synthesis occurs in dense clusters (Fig. 2D and 7B). A recent in vitro study using Xenopus extracts implied that replication

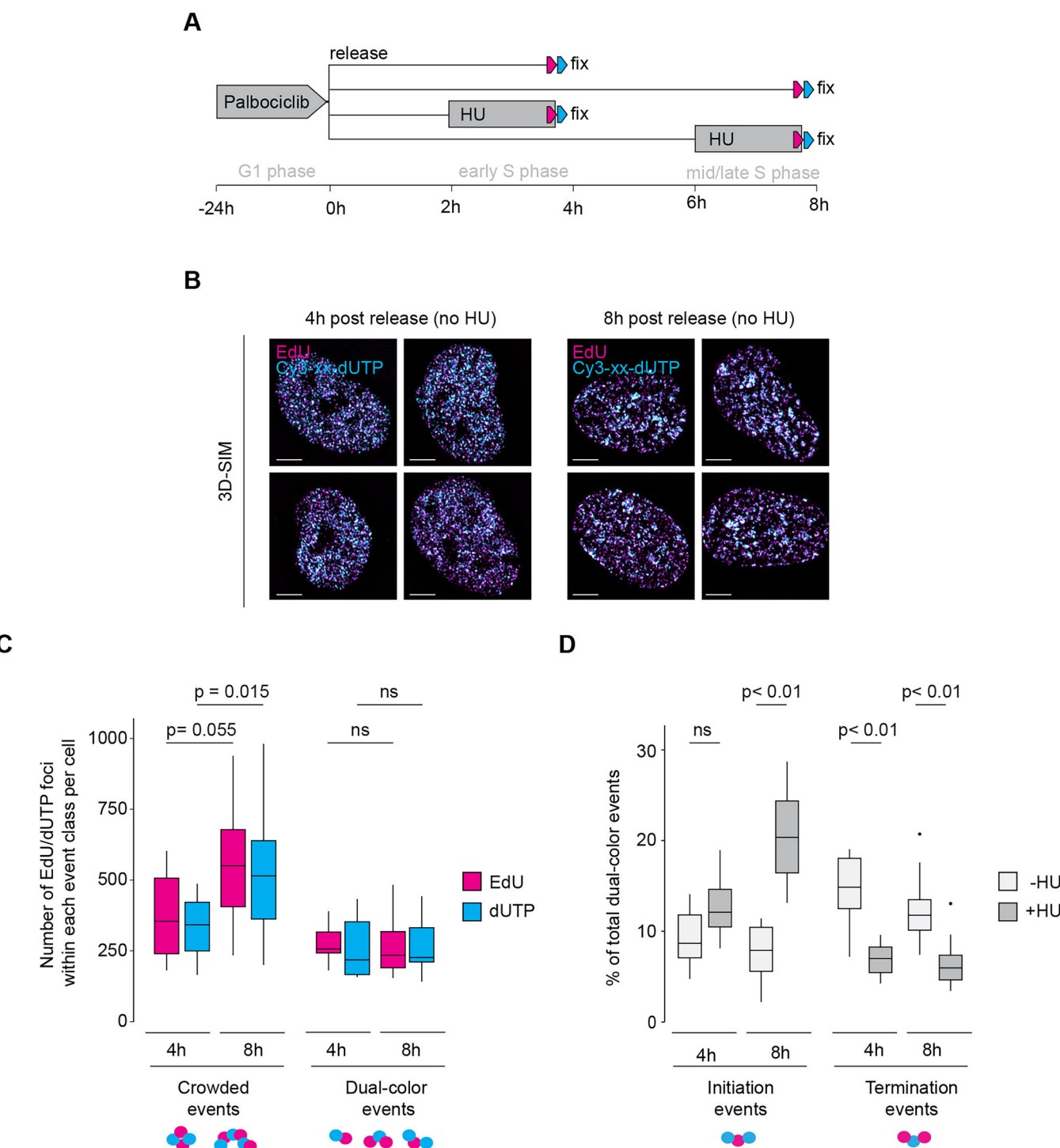

dynamics are not linear during S-phase and found PLK1 to direct different DNA replication modes (Ciardo et al, 2025). While here we focused on early and mid S-phase cells to develop 3D-SPARK, we have previously identified a causal link between DNA replication levels in late S-phase and nuclear PLK1 activity (Lemmens et al, 2018). Further studies are needed to define DNA replication dynamics in late S-phase cells and determine if human PLK1 controls DNA replication progression in 3D. Recent

innovations in whole-tissue imaging and super-resolution microscopy allow examination of entire organs at subcellular resolution or single molecules with precisions in the single-digit nanometre range (Takei, 2023; Sahl et al, 2024; Schueder et al, 2024).

Advances in single-molecule localisation microscopy (SMLM) techniques (Fu et al, 2023; Geertsema et al, 2021; Li et al, 2021; Ostersehlt et al, 2022) will help resolving dense foci clusters like those found in late-replicating domains. While high-resolution

**Figure 7. Analysis of DNA replication dynamics in synchronised S-phase cells.**

(A) Treatment scheme to obtain synchronised RPE1 cells for time-resolved 3D-SPARK; grey HU block indicates a 2-h pulse of 2 mM hydroxyurea. Magenta and Blue arrows indicate the standard pulsing scheme akin Fig. 2E. (B) Four representative 3D-SIM max projection images of synchronised RPE1 cells at indicated timepoints; scale bars = 5 μm. Number of nuclei analysed: 10 (4 h, mock); 12 (4 h, HU); 13 (8 h, mock); 12 (8 h, HU). (C) Box plot showing the number of EdU (magenta) or Cy3-xx-dUTP foci (blue) per nucleus within each event type depicted under the graph. RPE1 were treated as depicted in A and at least ten cells per condition were analysed. Number of nuclei: 10 (4 h, mock); 12 (4 h, HU); 13 (8 h, mock); 12 (8 h, HU). For each box, the inner line indicates median and box limits show 25th and 75th percentiles. Whiskers extend to edge values within 1.5 times the interquartile range between 25th and 75th percentiles from the box limits. Dots represent values beyond whisker range. P values were calculated with Mann–Whitney U-test and adjusted for multiple testing using Bonferroni correction. (D) Box plot showing the relative frequency of initiation and termination events per nucleus among segmented dual-colour events in cells treated according to the scheme in (A). Per condition more than ten synchronised cells were analysed, each containing >100 individual dual-colour events. Number of nuclei: 10 (4 h, mock); 12 (4 h, HU); 13 (8 h, mock); 12 (8 h, HU). For each box, the inner line indicates median and box limits show 25th and 75th percentiles. Whiskers extend to edge values within 1.5 times the interquartile range between 25th and 75th percentiles from the box limits. Dots represent values beyond whisker range. P values were calculated with Mann–Whitney U-test and adjusted for multiple testing using Bonferroni correction; $p = 1.1e\text{-}5$ (8 h, initiation, mock vs HU); $p = 2.6e\text{-}3$ (4 h, termination, mock vs HU); $p = 2.9e\text{-}3$ (8 h, termination, mock vs HU). Source data are available online for this figure.

imaging of single replication foci will provide much-needed structural insights into the geometry of human replication forks and chromatin assemblies, we believe numerous valuable insights can be gained from analysing DNA replication at ~100 nanometre spatial resolution and directly relating endogenous biomarkers with two quantitative nascent DNA markers in the same human cell—a feat well performed by 3D-SIM and ExM. Incorporating the recent advances of SMLM into the 3D-SPARK approach will further improve its versatility, accuracy and throughput capacity, which ultimately could provide a 3D view on DNA replication across scales and thus help researchers to bridge the gap between structural biology, 2D fibres, genomics and human cell biology. The flexibility and accessibility of our 3D-SPARK approach will facilitate exploration of the spatial component of DNA replication regulation and directly connect cell status to DNA nanostructure within a single image.

# Methods

**Reagents and tools table**

| Reagent/resource | Reference or source | Identifier or catalogue number |
|---|---|---|
| **Experimental models** | | |
| hTERT-immortalised RPE1 cells | ATCC | CRL-4000 |
| HBEC CDC6 Tet-On cells | Komseli et al, 2018 | N/A |
| **Antibodies** | | |
| Phospho-histone H2A.X [Ser139] | Cell Signaling Technology | 2577 |
| Phospho-RPA32 [S33] | Bethyl Laboratories | A300-246A |
| PCNA (PC10) | Santa Cruz Biotechnology | sc-56 |
| AlexaFluor 488 anti-rabbit | Life Technologies | A11008 |
| **Oligonucleotides** | | |
| RIF1 esiRNA | Sigma-Aldrich | EHU095481-20UG |
| GFP esiRNA | Sigma-Aldrich | EHUEGFP-20UG |
| **Chemicals, enzymes and other reagents** | | |
| Hydroxyurea | Sigma-Aldrich | H8627 |
| Doxycycline | Sigma-Aldrich | D9881 |
| Paraformaldehyde, 4% | Histolab | 02176 |

| Reagent/resource | Reference or source | Identifier or catalogue number |
|---|---|---|
| Methanol, cold | Sigma-Aldrich | N/A |
| BSA | Sigma-Aldrich | A9418 |
| 5-ethynyl-2′-deoxyuridine (EdU) | Jena Bioscience | CLK-N001-100 |
| 5-ethynyl-arabino-uridine (F-ara-EdU) | Jena Bioscience | CLK-1403-25 |
| Alexa Fluor 647 Azide | Invitrogen | A10277 |
| Tris Base | Sigma-Aldrich | T6066 |
| $CuSO_4$ | Sigma-Aldrich | C1297 |
| Ascorbic acid | Sigma-Aldrich | A4544 |
| SNTT1 | Sigma-Aldrich | SCT064 |
| Cy3-dUTP | Jena Bioscience | NU-803-CY3-S |
| Cy3-xx-dUTP | Jena Bioscience | NU-803-XX-CY3-S |
| Cy5-xx-dUTP | Jena Bioscience | NU-803-XX-CY5-S |
| dUTP-XX-STAR RED | Jena Bioscience | NU-803-XX-STRED-S |
| Tricine buffer | Kužmová et al, 2021 | N/A |
| DPBS | Gibco | 2037539 |
| TBS/T | N/A | N/A |
| DAPI | Thermo Fisher Scientific | D1306 |
| Poly-L-lysine | Sigma-Aldrich | A-005-C |
| Acrylamide | Sigma-Aldrich | A4058-100ML |
| Formaldehyde | Sigma-Aldrich | 252549-500 ML |
| Sodium acrylate | Sigma-Aldrich | 408220-25 G |
| N,N′-Methylenebisacrylamide | Sigma-Aldrich | M1533-25ML |
| Ammonium persulphate | Sigma-Aldrich | A3678-25G |
| N,N,N′,N′-Tetramethylethylenediamine | Sigma-Aldrich | T7024-25ML |
| Sodium dodecyl sulphate | Sigma-Aldrich | 74255 |
| Sodium chloride | Thermo Fisher Scientific | S73161/60 |
| UltraPure Low Melting Point Agarose | Thermo Fisher Scientific | 16520050 |
| Ibidi glass bottom imaging dishes | Ibidi | 81158 |

| Reagent/resource | Reference or source | Identifier or catalogue number |
|---|---|---|
| **Software** | | |
| CellProfiler | Broad Institute | N/A |
| R (statistical software) | R Foundation | N/A |
| Adobe Illustrator | Adobe | N/A |
| ImageJ | NIH | N/A |
| PyClesperanto | Haase et al, 2023 | N/A |
| skimage (scikit-image) | Gouillart et al, 2016 | N/A |
| Huygens deconvolution software | Scientific Volume Imaging | N/A |
| **Other** | | |
| DMEM-F12 GlutaMAX | Thermo Fisher Scientific | 31331093 |
| Penicillin-Streptomycin | Thermo Fisher Scientific | 15140122 |
| Foetal Bovine Serum (FBS), heat-inactivated | Sigma-Aldrich | F7524 |
| Keratinocyte-SFM Medium | Thermo Fisher Scientific | 10144892 |
| EGF | Thermo Fisher Scientific | 10144892 |
| Bovine Pituitary Extract (BPE) | Thermo Fisher Scientific | 10144892 |
| Opti-MEM | Thermo Fisher Scientific | 31985062 |
| Lipofectamine RNAiMAX | Thermo Fisher Scientific | 13778150 |

## Cell culture

Human hTERT-immortalised retinal pigment epithelial cells (hereafter referred to as RPE1) were obtained from the American Type Culture Collection (ATCC, CRL-4000) and cultured in an ambient-controlled incubator at 37 °C and 5% $CO_2$ and maintained using DMEM-F12 GlutaMAX (Thermo Fisher Scientific, 31331093) supplemented with 1% penicillin-streptomycin (Thermo Fisher Scientific, 15140122) and 10% heat-inactivated foetal bovine serum (Sigma-Aldrich, F7524). The p53 deficient RPE1 cells were cultured the same as parental RPE1 cells and are characterised previously (Cascales et al, 2021). Human bronchial epithelial cells (hereafter referred to as HBEC) CDC6 Tet-On cells, were previously described (Komseli et al, 2018) and cultured in Keratinocyte-SFM Medium with 5 ng/ml EGF and 50 μg/ml bovine pituitary extract (BPE) (Thermo Fisher Scientific, 10144892), supplemented with 1% penicillin-streptomycin (Thermo Fisher Scientific, 15140122). All cell lines were authenticated using SNP mapping, IF and biochemistry methods within the last 3 years and regularly confirmed to be mycoplasma-free.

## Drug treatments and esiRNA transfection

RPE1 cells were seeded on glass cover slides (Thermo Fisher Scientific, 11846933) placed in 12-well imaging plates (Sardtedt,

83.3921) or in 96-well imaging plates (Sigma-Aldrich, CLS3904-100EA), for 24 h prior to treatment. The following small molecule drugs were used at the indicated final concentrations, unless otherwise specified: hydroxyurea (2 mM, Sigma-Aldrich, H8627), doxycycline (1 μg/ml, Sigma-Aldrich, D9881). Mock-treated controls were exposed to an equivalent volume of media, without any active compound. For esiRNA transfection, a master mix was prepared by combining Opti-MEM (Thermo Fisher, 31985062) with Lipofectamine RNAiMAX (5.6 μl/ml, ThermoFisher, 13778150). A separate mix of RNAi was prepared by combining RIF1 esiRNA (20 nM, Sigma-Aldrich, EHU095481-20UG) and GFP esiRNA (20 nM, Sigma-Aldrich, EHUEGFP-20UG). The master mix and RNAi mix were combined before being added to cells (final concentration of both esiRNAs 21.3 ng/μl), which were incubated for 48-hours with the esiRNA.

## Nascent DNA labelling

An illustrated step-by-step protocol for dual colour nascent DNA labelling is provided as an Appendix Protocol. For nucleoside pulse labelling, 5-ethynyl-2′-deoxyuridine (EdU) (Jena Bioscience, CLK-N001-100) or 5-ethynyl-arabino-uridine (F-ara-EdU) (Jena Bioscience, CLK-1403-25) was added to the media of live cells at the stated concentration and for the stated amount of time. For fluorescent dNTP pulse labelling, Cy3-dUTP, Cy3-xx-dUTP, Cy5-dUTP or Cy5-xx-dUTP (Jena Bioscience, NU-803-CY3-S, NU-803-XX-CY3-S, NU-803-XX-CY5-S, where 'xx' denotes a linker structure between the nucleoside and fluorophore molecule), were mixed with a synthetic nucleoside triphosphate transporter (SNTT1) (6 μM, Sigma-Aldrich, SCT064) as follows: the dNTP and the transporter were diluted in tricine buffer (Kužmová et al, 2021), each diluted component was mixed 1:1 to form the transfection mixture. Media was aspirated from the cells, followed by incubation with the transfection mixture for the stated amount of time to allow uptake of the fluorescent dNTP (pulse). The mixture was then removed, the cells were washed with media and incubated with conditioned media for the stated amount of time to allow nucleotide incorporation (chase). For example, a 15-min pulse consists of a 4-min transfection pulse and an 11-min chase, written as '15 (4 + 11) min'. For simultaneous Cy3-xx-dUTP and Cy5-xx-dUTP pulse labelling, two dNTP/SNTT1 mixes were prepared separately and then combined for the final transfection mixture of desired dNTP concentrations. The standard pulsing scheme is defined as a 15-min EdU pulse followed by a 15 (4 + 11)-min Cy3-xx-dUTP treatment. After treatments, cells were washed in DPBS (Gibco, #2037539), fixed in 4% paraformaldehyde solution (Histolab, 02176) for 7 min, permeabilised in cold methanol (Sigma Aldrich) for 2 min, washed again in DPBS and blocked with 2% BSA (Sigma-Aldrich A9418) in TBS/T (i.e. Tris buffered saline solution supplemented with 0.1% Tween-20). If immunostaining applied, fixed samples were incubated overnight at 4 °C with a primary antibody: phospho-histone H2A.X [Ser139] (1:800, Cell Signalling Technology, #2577); phospho-RPA32 [S33] (1:2000, Bethyl Laboratories, #A300-246A); or PCNA (PC10) (1:400, Santa Cruz, sc-56) in blocking media. Samples were washed in TBS/T and DPBS, then incubated with a secondary antibody AlexaFluor 488 anti-rabbit (1:800, Life Technologies, #A11008) and 50 ng/ml DAPI (Thermo Fisher Scientific, D1306) for 1 h at room temperature. Samples were washed again in TBS/T and DPBS, and EdU click

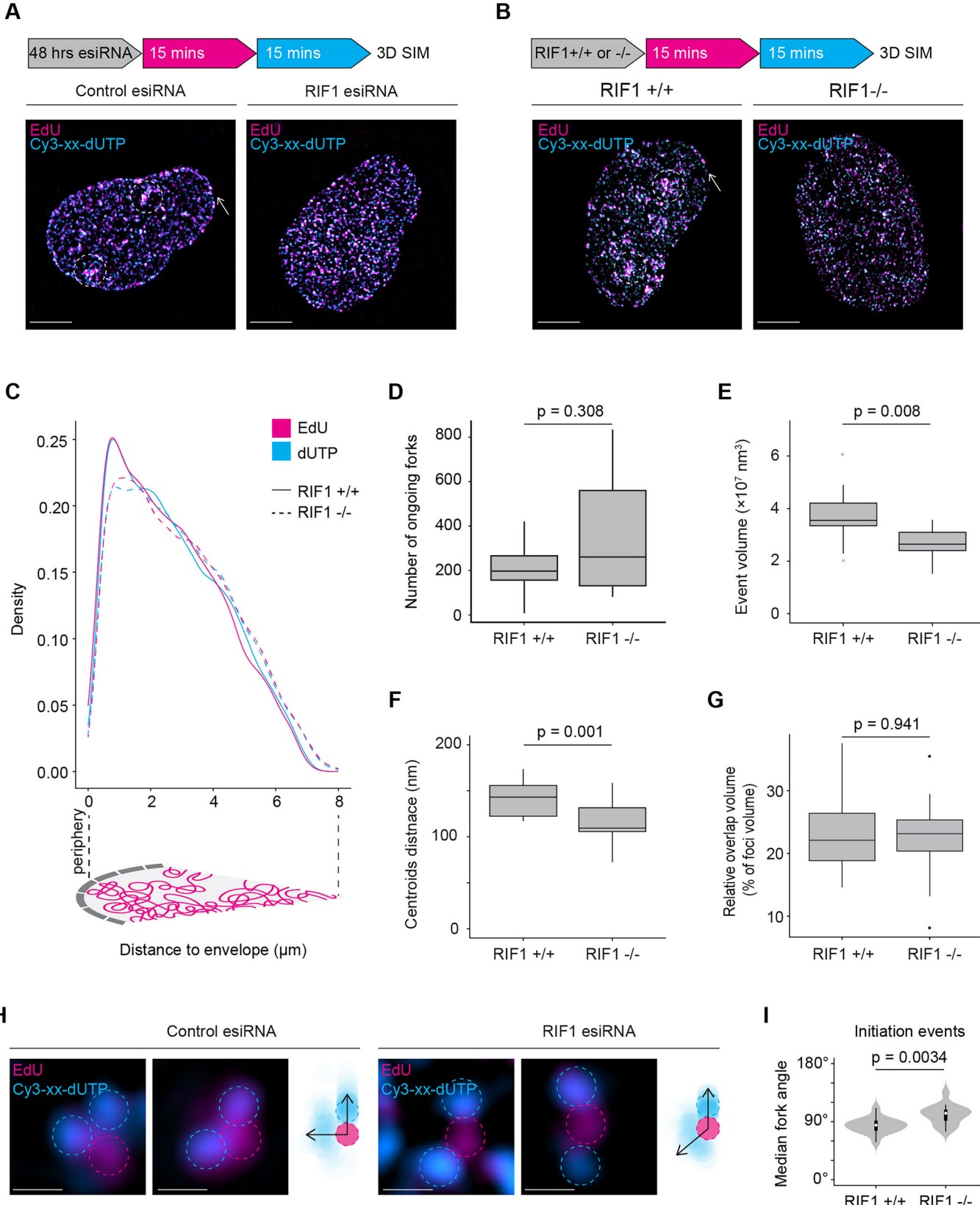

**Figure 8.   RIF1 loss changes the nuclear location and 3D constellation of replication nanostructures.**

(A) Representative 3D-SIM images of cells treated for 48 h with GFP esiRNA (control) or RIF1 esiRNA before labelling with the standard pulsing scheme, scale bars = 5 µm. Dashed circles indicate regions with a high density of replication events. The arrow highlights enriched replication events at the nuclear periphery. (B) Representative 3D-SIM images of RIF1 +/+ (n = 17 cells) and RIF1 −/− RPE1 cells (n = 15 cells), treated with the standard pulsing scheme, scale bars = 5 µm. Dashed circles indicate regions with a high density of replication events. The arrow highlights enriched replication events at the nuclear periphery. Density plots describing the distribution of replication events relative to their distance from the nuclear periphery in RIF1 +/+ and RIF1 −/− cells. (C) Density plots describing the distribution of foci (>50 voxels) relative to their distance from the nuclear periphery in RIF1 +/+ and RIF1 −/− cells. (D) Box plot showing the total number of *ongoing events* per nucleus in RIF1 +/+ and RIF1 −/− cells. For each box, the inner line indicates median and box limits show 25th and 75th percentiles. Whiskers extend to edge values within 1.5 times the interquartile range between 25th and 75th percentiles from the box limits. Dots represent values beyond whisker range. *P* values were calculated with Mann–Whitney *U*-test and adjusted for multiple testing using Bonferroni correction. (E) Box plot showing the volume of ongoing events in RIF1 +/+ and RIF1 −/− cells. For each box, the inner line indicates median and box limits show 25th and 75th percentiles. Whiskers extend to edge values within 1.5 times the interquartile range between 25th and 75th percentiles from the box limits. Dots represent values beyond whisker range. *P* values were calculated with Mann–Whitney *U*-test and adjusted for multiple testing using Bonferroni correction. (F) Box plot showing the centroid distances between EdU and Cy3-xx-dUTP foci of ongoing events in RIF1 +/+ and RIF1 −/− cells. For each box, the inner line indicates median and box limits show 25th and 75th percentiles. Whiskers extend to edge values within 1.5 times the interquartile range between 25th and 75th percentiles from the box limits. Dots represent values beyond whisker range. *P* values were calculated with Mann–Whitney *U*-test and adjusted for multiple testing using Bonferroni correction. (G) Box plot showing the ratio of overlapping EdU and Cy3-xx-dUTP foci volume versus the total event volume of ongoing events in RIF1 +/+ and RIF1 −/− cells. For each box, the inner line indicates median and box limits show 25th and 75th percentiles. Whiskers extend to edge values within 1.5 times the interquartile range between 25th and 75th percentiles from the box limits. Dots represent values beyond whisker range. *P* values were calculated with Mann–Whitney *U*-test and adjusted for multiple testing using Bonferroni correction. (H) Representative 2D maximum projections of initiation events of cells treated according to 1A, scale bar: 250 nm. Estimated positions of EdU foci centres (magenta dashed circle) and Cy3-xx-dUTP foci centres (cyan dashed circle) were annotated on the images and used to generate a transparency density plot, depicted on the right, n = 73, control esiRNA and n = 72, RIF1 esiRNA. The arrows indicate the estimated position of the highest Cy3-xx-dUTP densities. (I) Violin/box plot showing the median 3D fork angle per cell between Cy3-xx-dUTP foci within initiation for RIF1 +/+ and RIF1 −/− cells treated according to 1B. The box shows the interquartile range (IQR) from 25th (Q1) to 75th (Q3) percentile; the white dot in the box corresponds to the median. Whiskers extend to the minimum and maximum values within 1.5 x IQR from Q1 and Q3. *P* values were calculated with Mann–Whitney *U*-test and adjusted for multiple testing using Bonferroni correction. Source data are available online for this figure.

chemistry was performed by incubation in 100 mM Tris Base (Sigma-Aldrich, T6066), 1 mM CuSO$_4$ (Sigma, C1297), 100 mM ascorbic acid (Sigma-Aldrich, A4544), and 0.5 µM Alexa Fluor 647 Azide (Invitrogen, A10277) for 1 h at room temperature, followed by further washes in TBS/T and DPBS. Stained samples were stored in DPBS.

## Quantitative imaged-based cytometry (QIBC)

Widefield images of large cell populations (>1000 cells/replicate condition) were acquired at room temperature using a Nikon Ti2 ECLIPSE microscope with a 20X air objective and analysed using custom CellProfiler and R pipelines (de Souza Gama et al, 2024), then visualised using Adobe Illustrator. CellProfiler analysis included a basic illumination correction using a smoothened image (Fit Polynomial) for all wavelengths and single nuclei were identified as primary objects based on DAPI, a typical object diameter between 6 and 40 pixels and adaptive Otsu thresholding. Detailed confocal images were obtained using a Nikon-Ti2 Eclipse imaging system with a Re-scan Confocal Microscopy module (RCM2, Confocal.nl).

## 3D structured illumination microscopy (3D-SIM)

A stepwise 3D-SPARK protocol and SIM sample preparations is provided as an Appendix Protocol. To prepare the samples for nanoscale imaging, RPE1 or HBEC cells were seeded on 18 mm, high-precision no. 1.5 glass coverslips (Marienfeld, 0111580). Drug treatments and DNA pulse labelling were performed as previously described. Following treatments, samples were fixed with 4% PFA (16% Paraformaldehyde Aqueous Solution, EM Grade—Electron Microscopy Sciences, 15710) and incubated with primary and secondary antibodies, DAPI and EdU click chemistry, as previously described. Samples were mounted on ProLong Gold Antifade

Mountant (Thermo Fisher Scientific, P10144), on SuperFrost Ultra Plus Adhesion Slides (Thermo Fisher, 1014356190) and stored at 4 °C.

Super-resolution 3D-SIM imaging was performed on a Carl Zeiss Elyra PS.1 microscope equipped with high performance filter cubes and 405-, 488-, 561- and 642-nm excitation lasers. Multi-colour images were sequentially captured onto an Andor iXon DU 897 camera using a Plan-Apochromat 100x/1.46 NA, oil objective. Emission was collected sequentially through appropriate dichroic mirrors and bandpass filters set at 495–575 nm for 488 nm excitation, 570–650 nm for 561 nm excitation and above 655 nm for 642 nm excitation. 3D-SIM processing was done with the integrated ELYRA S system software (Zen 2011 SP2 Black), with a selection of automatic settings for evaluation of the raw data (i.e. theoretical PSF, selection of noise filter setting, frequency weighting, baseline settings, etc.). The optimal grid size was automatically assigned to each wavelength by the Zeiss Zen software, and the grid was rotated five times at five phases for each image. Calibration on 40 nm sub-resolution beads generated a lateral precision of about 85 ± 5 nm with 488 nm excitation as shown before (Agostinho et al, 2016).

### 3D-SIM image analysis

Image segmentation of reconstructed 3D-SIM stacks was performed using custom pipelines in Python. The used segmentation and image analysis pipelines are available via the public repository: https://github.com/LemmensLab/3D-SPARK. In brief, masks indicating the location of selected nuclei were generated manually based on maximum intensity Z-projections of the stacks. For each selected nucleus, the 3D-stack was anisotropy corrected according to Bio-Image Analysis Notebooks (Haase et al, 2024), to improve equal scaling across dimensions. Each channel was further segmented via a 3D Voronoi-Otsu-labelling workflow (combining Gaussian blur, spot detection, thresholding and binary

watershedding) with the PyClesperanto package (Haase et al, 2023). Segmented foci outside the nuclear mask were discarded from further analysis. Within the masks, a size cutoff was applied to exclude small objects (<50 voxels) classified as too small, to direct the downstream analysis on in-focus signals. The relative location and overlap of segmented foci were compared across channels, using the skimage package (Gouillart et al, 2016). For each segmented focus that has multiple overlaps, each overlap was scored according to their size relative to the biggest overlap of that same focus. Overlaps with sizes >20% of the maximum overlap for each focus were selected for further event classification, effectively prioritising key overlaps in crowded environments.

Remaining foci were classified as the following event types. Isolated foci that have no overlaps were classified as single colour events. Dual colour events, include three subclasses that were defined as: a single EdU region overlapping with a single dUTP region formed an ongoing event, a single EdU region overlapping with two dUTP regions that only overlapped with this one EdU region formed an initiation event, a single dUTP region overlapping with two EdU regions that only overlapped with this one dUTP region formed a termination event. Remaining events that do not fulfil these definitions are labelled as crowded and were not used in downstream analysis.

For centroid distance analysis of ongoing events, we measured the Euclidean distance between the centroid of the two foci that compose an ongoing event. For 3D geometry analysis of initiation events, the angles between two vectors going from the EdU centroid to each dUTP centroid were calculated as the arccosine of the two vectors. Resulting topological event type metrics for each nucleus were imported into the analysis programme R for comparison between images and conditions, and for plotting the data. ImageJ software was used for multi-colour image representation, rolling-ball background subtraction and 2D maximum projections. Signal intensity profiles of individual replication events were plotted using ImageJ and visualised using Adobe Illustrator.

## DNA fibre assays

RPE1 cells were seeded in six-well imaging plates (Sardtedt, 83.3920) and incubated for 24-h prior to treatment. DNA pulse labelling was performed for 15 $(4 + 11)$-min using Cy3-xx-dUTP and Cy5-xx-dUTP (1:1) as described above. Following fluorescent nucleotide labelling, 2000 cells were lysed directly onto SuperFrost Plus Adhesion Microscope Slides (Epredia, 11300), and DNA fibres were stretched by inverting the slides, as previously described (Jackson and Pombo, 1998). The fibres were allowed to air dry before fixation with cold methanol (Sigma-Aldrich, 32213-2.5L-M) and acetic acid (Sigma-Aldrich, 45726-1LF) (ratio, 3:1) on ice for 10 min. After drying, the fixed slides were stored at 4 °C. Slides were washed with dH$_2$O and denatured in 2.5 M HCl (Merck, 100317) for 45 min. After denaturation, the slides were sequentially washed in DPBS, incubated in DPBS with 1% Triton X-100 (Sigma-Aldrich, T8787; hereafter referred to as '1% Triton') for 5 min, washed again DPBS, washed in DBPS containing 1% BSA (Sigma-Aldrich, A7906), 0.1% Tween-20 (Sigma-Aldrich, P7949), and 100 µL 10% sodium azide (Sigma-Aldrich, S2002; hereafter referred to as 'DPBS+') and incubated in DPBS+ for 15 min. The slides were incubated in DPBS for 15 min, washed twice in DPBS+, and washed twice in DPBS. 4% formaldehyde solution (Histolab, 02176) was added to the slides for 10 min at room

temperature, followed by washing the slides twice with DPBS, twice with DPBS+, and once with dH$_2$O. After incubation overnight, the slides were washed twice in DPBS+ and once in dH$_2$O. Finally, the slides were mounted using ProLong Gold Antifade Mountant (Thermo Fisher Scientific, P10144) and stored at 4 °C. The fluorescent tracts were imaged using a Nikon-Ti2 Eclipse imaging system with a re-scan confocal microscopy module (RCM2, Confocal.nl) and analysed by manual counting with ImageJ. For each slide, >450 Cy-3-positive forks were scored, and measurements of three independent slides were pooled to determine tract size distribution. Data were visualised using custom R pipelines and Adobe Illustrator.

## Expansion microscopy-based assay for replication kinetics (XMARK)

A stepwise protocol for nascent DNA labelling and XMARK is provided as an Appendix Protocol. XMARK was performed with a variant of the original magnified analysis of the proteome (MAP) protocol., as described in (Zwettler, Reinhard, et al, 2020). In brief, RPE1 cells were seeded on poly-L-lysine-coated (Sigma-Aldrich, A-005-C) 12-mm glass coverslips (Thermo Fisher Scientific, 1014356190) placed in 12-well plates (Sarstedt, 83.3921). DNA pulse labelling was performed as described above with EdU for 30-min followed by a 20 $(4 + 16)$-min pulse with Aminoallyl-dUTP-XX-STAR RED (6 µM, Jena Bioscience, NU-803-XX-STRED-S). Coverslips were washed with DPBS and fixed in 4% paraformalde-hyde solution (Histolab, 02176) for 15 min, permeabilised in cold methanol (Sigma-Aldrich) for 2-min and washed again in DPBS. After fixation, proteins were anchored with 30% acrylamide (Sigma-Aldrich, A4058-100ML), 4% formaldehyde (Sigma-Aldrich, 252549-500 ML) and DPBS for 4 h at 37 °C and washed in DPBS. Crosslinking polymerisation of hydrogels was performed on ice with the following monomer solution: 7% sodium acrylate (Sigma-Aldrich, 408220-25 G), 20% acrylamide (Sigma-Aldrich, A4058-100ML), 0.05% $N,N'$-methylenebisacrylamide (Sigma-Aldrich, M1533-25ML), 0.5% ammonium persulphate (Sigma-Aldrich, A3678-25G), 0.5% $N,N,N',N'$-tetramethylethylenediamine) (Sigma-Aldrich, T7024-25ML) in DPBS for 1.5 h at 37 °C. Polymerised hydrogels were denatured in 200 nM sodium dodecyl sulphate (Sigma-Aldrich, 74255), 200 mM sodium chloride (Thermo Fisher Scientific, S73161/60) and 50 mM Tris base (Sigma-Aldrich, T6066-1KG) at pH9.0 for 1 h at 95 °C. Samples were swelled in dH$_2$0, exchanged several times until maximum expansion was achieved.

EdU-click labelling was performed post-expansion. Fully expanded gels were cut to an appropriate size and EdU-click labelling was performed by incubating in 100 mM Tris (Sigma-Aldrich, T6066-1KG), 1 mM CuSO$_4$ (Sigma, C1297), 100 mM ascorbic acid (Sigma-Aldrich, A4544), and a fluorescent dye azide (Invitrogen, A10277) diluted in dH$_2$0, for 1 h at room temperature, followed by further washes in DPBS. Gels were then stained with 500 ng/ml DAPI for 1.5 h at 37 °C and washed in DPBS. Samples were re-expanded in dH$_2$0 and mounted on poly-L-lysine coated 35 mm, glass bottom imaging dishes (Ibidi, 81158), with 1% UltraPure Low Melting Point Agarose (Thermo Fisher Scientific, 16520050) and with dH$_2$0 added on top to prevent shrinkage during image acquisition.

Expanded gels with fluorescently labelled EdU were imaged using a Zeiss LSM780 confocal microscope (with Zen black 2012

SP5 software) and a Zeiss Plan-Apochromat 40x/1.2NA, water objective; x–y pixel size, 55.0 nm; z-step size, 0.4–0.5 μm; pixel dwell time, 0.42 μs. The excitation wavelenghts used were 405 and 488 nm, with emission filters set at 409–481 nm and 499–561 nm, respectively. Non-expanded RPE1 cells were imaged on the same confocal microscope using a Zeiss Plan-Apochromat 63x/1.4 NA, oil objective; x–y pixel size, 26.3 nm; z-step size, 0.13 μm; line average, 2; pixel dwell time, 0.64 μs. Confocal images were deconvolved using Huygens deconvolution software (Professional 24.04 from Scientific Volume Imaging).

Expanded dual-colour labelled gels were imaged on a Zeiss 980 Airyscan2, with a Zeiss Plan-Apochromat 40x/1.2 NA, water objective. Excitation wavelengths used were 405, 488 and 639 nm, and emission passed through selected bandpass filter and onto the Airy array detector; x–y pixel size, 41 × 41 nm; z-step size, 0.5 μm; pixel dwell time, 0.35 μs. High-resolution images were processed using standard settings for sharpness using the fast Sheppard Sum mode in the Zen blue 3.10 software). Standard ImageJ tools were used for multi-colour image representation, processed with rolling-ball background subtraction, minor Gaussian blur ($\sigma = 2$) and maximum imaging projections. Signal intensity profiles of individual replication events were estimated using ImageJ and final figure galleries were assembled using Adobe Illustrator.

## Data availability

The code used for image preprocessing, segmentation, and quantitative analysis is openly accessible via GitHub at https://github.com/LemmensLab/3D-SPARK. The original microscopy images and all processed data supporting the findings of this study are available from the corresponding author upon reasonable request.

The source data of this paper are collected in the following database record: biostudies:S-SCDT-10_1038-S44318-025-00574-2.

## Peer review information

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

## Acknowledgements

We thank the Advanced Light Microscopy Facility (SciLifeLab, KTH Royal Institute of Technology) and Bioimage Informatics Facility (SciLifeLab, Uppsala University) for the guidance on sample processing and image analysis, and Dr. Apolinar Maya-Mendoza (Danish Cancer Society) for support on DNA fibre assays. We are grateful to Ann-Sofie Nilsson and Abid Sayyid for technical support and Dr. Arne Lindqvist (Karolinska Institutet) for helpful comments on the manuscript and for providing the RPE1 p53KO cell line and Dr. Steven P. Jackson (University of Cambridge) for gifting the RIF1-deficient RPE1 cells and parental controls. This research was made possible by an ASPIRE Award from The Mark Foundation for Cancer Research (Grant No. 23-040-ASP). We gratefully acknowledge funding from the Swedish Research Council (Starting grant 2019-04667; project grant VR-MH 2014-46602-117891-30), Jeanssons Foundations (grant 4-3007/2020), Åke Wiberg Foundation (Medicine grant M20-0066), Cancerfonden (Postdoc grant 170084), Karolinska Institutet (Assistant Professor Faculty grant 2-1534/2020; KID doctoral education grant; CRKI Blue Sky grant for innovative cancer research & technology), Boehringer Ingelheim Fonds (travel grant) and SciLifeLab (RED project grant).

## Author contributions

**Michael Hawgood**: Data curation; Software; Formal analysis; Investigation; Methodology; Writing—original draft; Writing—review and editing. **Bruno Urien**: Data curation; Software; Formal analysis; Investigation; Methodology; Writing—review and editing. **Ana Agostinho**: Investigation; Methodology;

Writing—review and editing. **Praghadhesh Thiagarajan**: Investigation; Methodology. **Giovanni Giglio**: Software; Formal analysis; Investigation. **Yiqiu Yang**: Formal analysis; Investigation. **Xue Zhang**: Formal analysis; Investigation. **Gemma Quijada**: Formal analysis; Investigation. **Matilde Fonseca**: Formal analysis; Investigation. **Jiri Bartek**: Funding acquisition; Writing—review and editing. **Hans Blom**: Methodology; Writing—review and editing. **Bennie Lemmens**: Conceptualisation; Resources; Data curation; Formal analysis; Supervision; Funding acquisition; Investigation; Visualisation; Methodology; Writing—original draft; Project administration; Writing—review and editing.

Source data underlying figure panels in this paper may have individual authorship assigned. Where available, figure panel/source data authorship is listed in the following database record: biostudies:S-SCDT-10_1038-S44318-025-00574-2.

## Funding

## Disclosure and competing interests statement

The authors declare no competing interests.

# Expanded View Figures

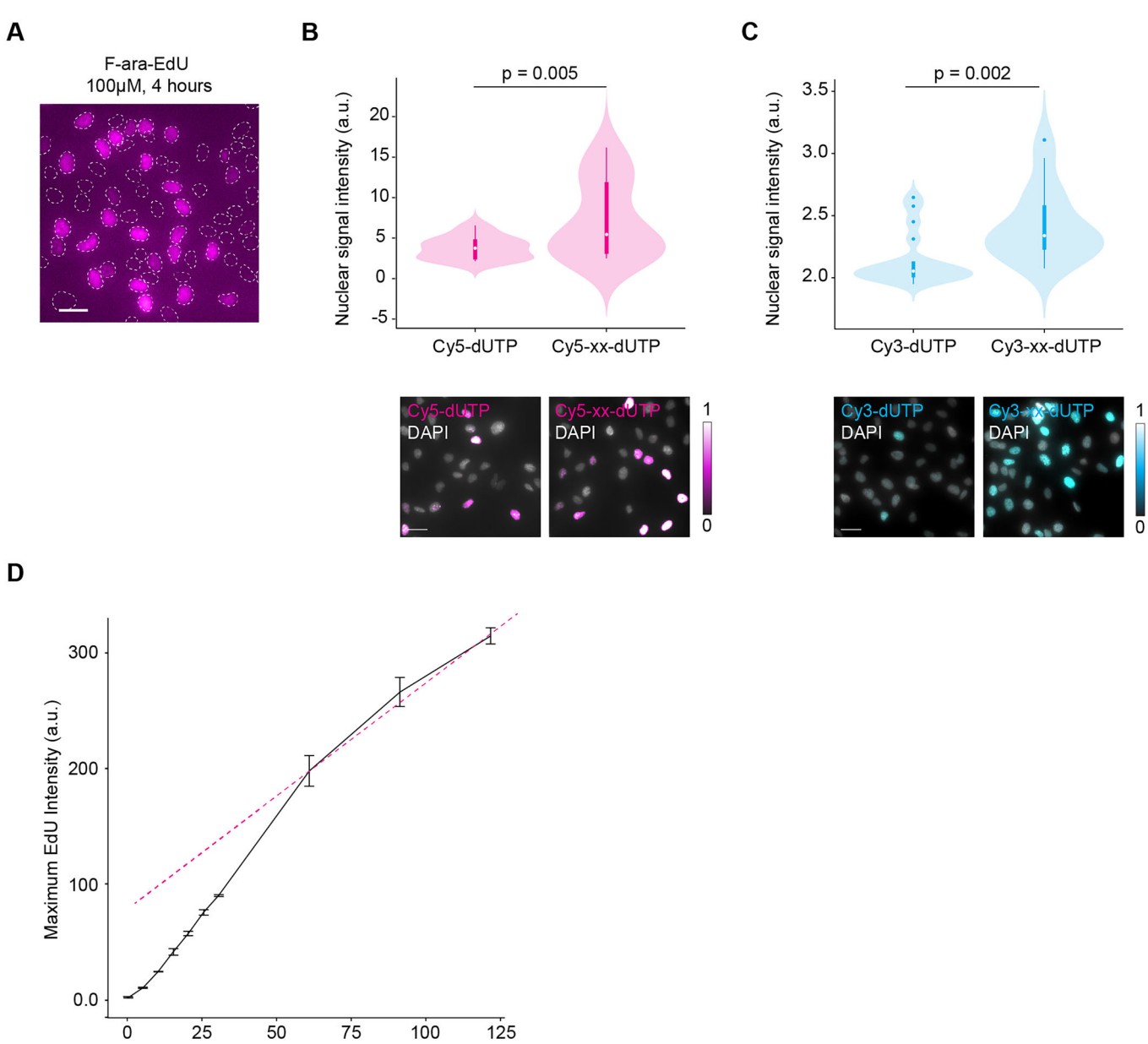

**Figure EV1. Additional characterisation of F-ara-EdU, fluorescent dUTP linker variants and EdU pulses.**

(A) Representative widefield fluorescence image after 4 h, 100 μM F-ara-EdU. Nuclei are annotated with a white dashed line; scale bar: 20 μm. (B) Violin/box plot showing the S phase nuclear signal intensities (a.u.) of Cy5-dUTP with and without an aminoallyl linker molecule (xx) of three independent experiments. Representative images are shown below, scale bar: 20 μm. For each box, the inner line indicates median and box limits show 25th and 75th percentiles. Whiskers extend to edge values within 1.5 times the interquartile range between 25th and 75th percentiles from the box limits. Dots represent values beyond whisker range. (C) Violin/box plot showing the S phase nuclear signal intensities (a.u.) of Cy3-dUTP with and without an aminoallyl linker molecule (xx) of three independent experiments. Representative images are shown below, scale bar: 20 μm. For each box, the inner line indicates median and box limits show 25th and 75th percentiles. Whiskers extend to edge values within 1.5 times the interquartile range between 25th and 75th percentiles from the box limits. Dots represent values beyond whisker range. (D) Line plot (black line) showing the average 95th percentile EdU intensity upon different EdU pulse durations (0, 5, 10, 15, 20, 25, 30, 90, 120 min). Cyan line highlights the linear trend of pulses ≥ 60 min, similar to the observation by Pereira et al Oncotarget 2017. Three replicates per condition; error bars represent SEM; At least 15,000 cells per conditions were analysed.

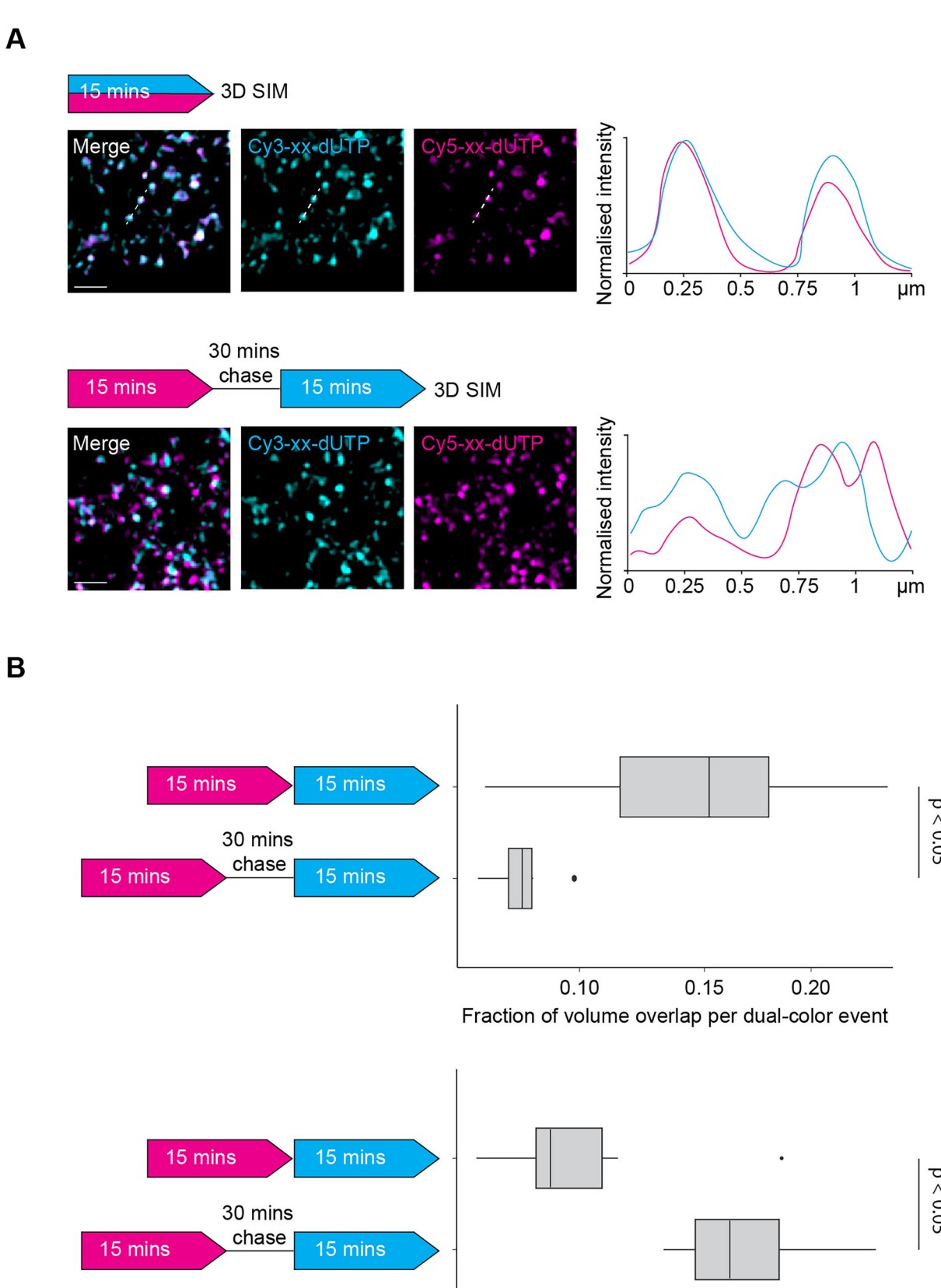

◀ **Figure EV2. Pulse-chase studies verifying replication foci dynamics detected by 3D-SPARK.**

(A) Representative 2D maximum projections (3D-SIM) of a nuclear region after a simultaneous pulse of Cy5-xx-dUTP and Cy3-xx-dUTP at a 1:1 ratio (upper panel) or after a 15-min pulse of Cy3-xx-dUTP, followed by a 30-min chase period and a final Cy5-xx-dUTP pulse. For each condition single wavelength and merged images are shown; scale bar = 1 µm. The line graph shows the normalised signal intensity profile across the depicted white dashed line. (B) Box plot reporting the fraction of volume overlap per dual-colour event (upper panel) and the distance between signal centroids (lower panel), following the EdU (magenta) and Cy3-xx-dUTP (blue) pulsing schemes indicated on the left: two consecutive pulses ($n = 6$ cells) or two pulses divided by a 30-min chase ($n = 7$ cells). For each box, the inner line indicates median and box limits show 25th and 75th percentiles. Whiskers extend to edge values within 1.5 times the interquartile range between 25th and 75th percentiles from the box limits. Dots represent values beyond whisker range. $P$ values were calculated with Mann–Whitney $U$-test and adjusted for multiple testing using Bonferroni correction.

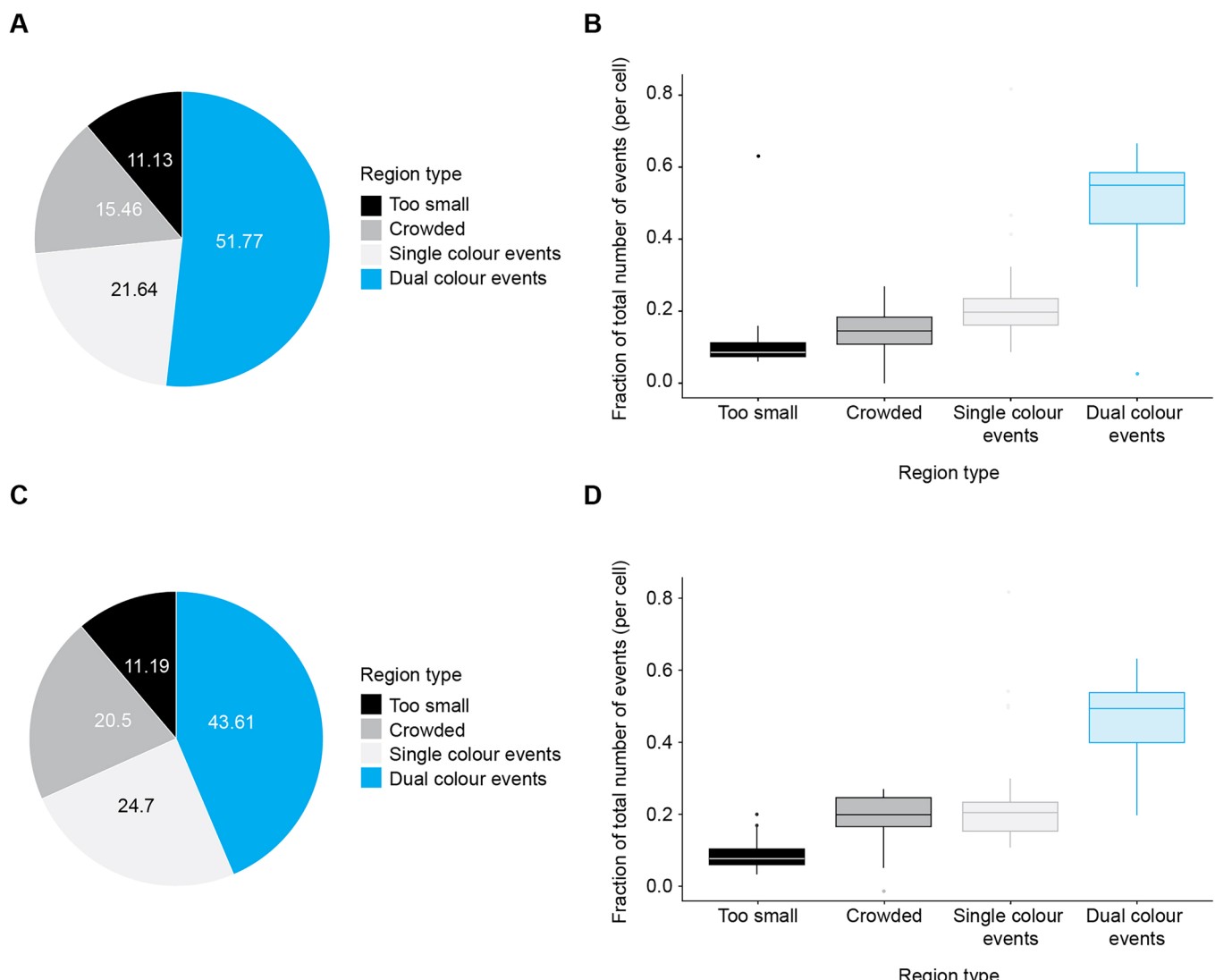

**Figure EV3.  Classification and relative proportions of fluorescent regions defined in image analysis.**

(**A**) Pie chart showing the percentage of different regions defined during image analysis of untreated early-S phase RPE1 cells ($n = 28$ cells). Regions that are defined as too small, *crowded* or single colour events, are removed from final analysis. Dual colour events, are used in the replication dynamics analysis. (**B**) Box plot showing the fraction of total number of events per RPE1 cell of different regions defined during image analysis. For each box, the inner line indicates median and box limits show 25th and 75th percentiles. Whiskers extend to edge values within 1.5 times the interquartile range between 25th and 75th percentiles from the box limits. Dots represent values beyond whisker range. (**C**) Pie chart showing the percentage of different regions defined during image analysis of untreated early-S phase RPE FRT/TR cells ($n = 17$ cells) (Chiang et al, 2016). Regions that are defined as too small, crowded or single colour events, are removed from final analysis. Dual colour events, are used in the replication dynamics analysis. (**D**) Box plot showing total number of dual colour events per cell in early-S phase RPE1 and RPE FRT/TR cells selected for replication dynamics analysis. For each box, the inner line indicates median and box limits show 25th and 75th percentiles. Whiskers extend to edge values within 1.5 times the interquartile range between 25th and 75th percentiles from the box limits. Dots represent values beyond whisker range.

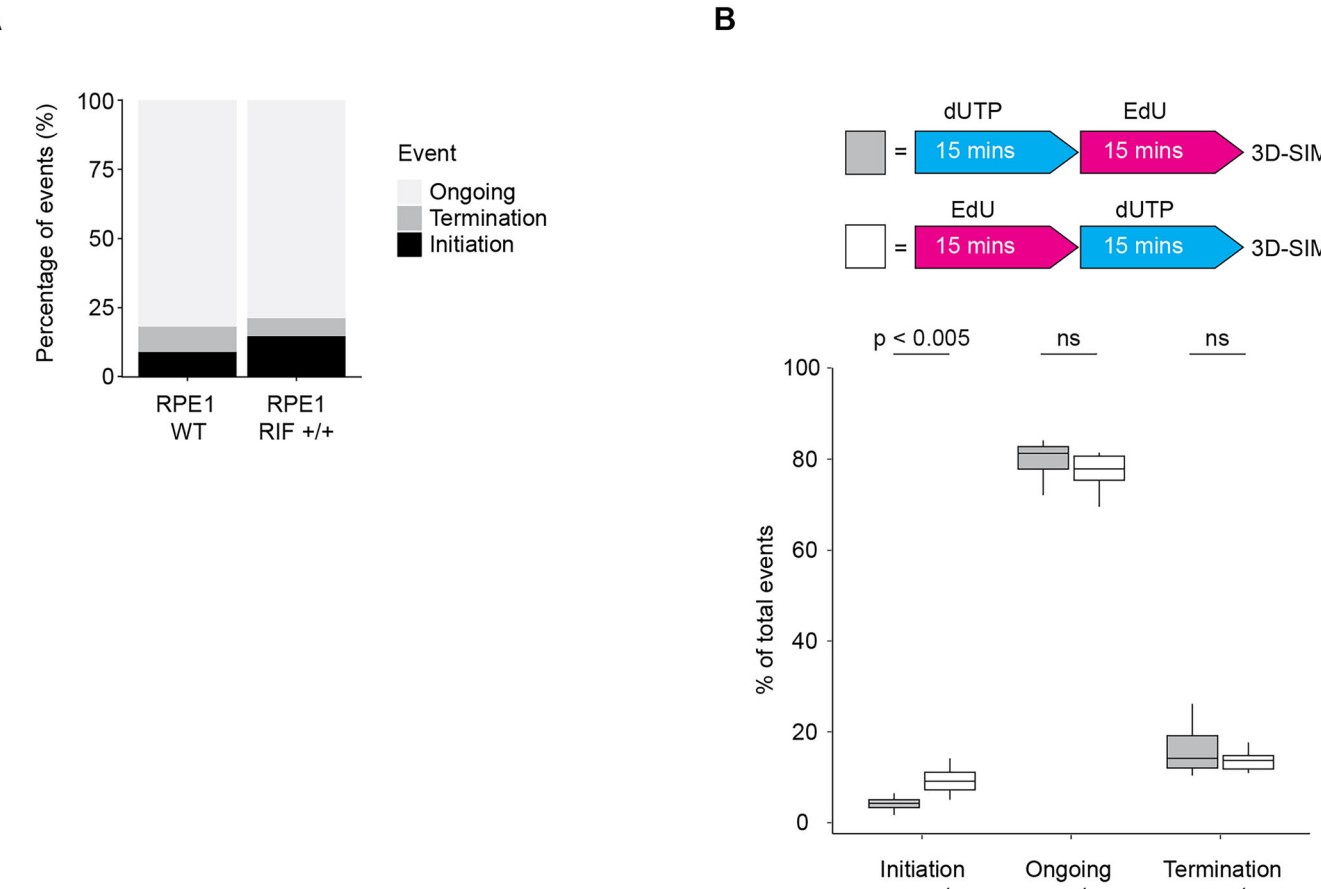

**Figure EV4. The proportion of different types of dual colour events.**

(A) Stacked bar graph shows relative percentage of ongoing, initiation and termination events in early-S phase RPE1 ($n = 10{,}390$ total number of dual colour events) and RPE FRT/TR cells ($n = 4957$ total number of dual colour events). (B) Box plot showing the percentage of event (initiation, ongoing, termination) on the total events, following the classic pulse scheme (EdU followed by dUTP, $n = 6$ cells) or the reversed one (dUTP followed by EdU, $n = 9$ cells). For each box, the inner line indicates median and box limits show 25th and 75th percentiles. Whiskers extend to edge values within 1.5 times the interquartile range between 25th and 75th percentiles from the box limits. Dots represent values beyond whisker range. P values were calculated with Mann–Whitney U-test and adjusted for multiple testing using Bonferroni correction.

**A**

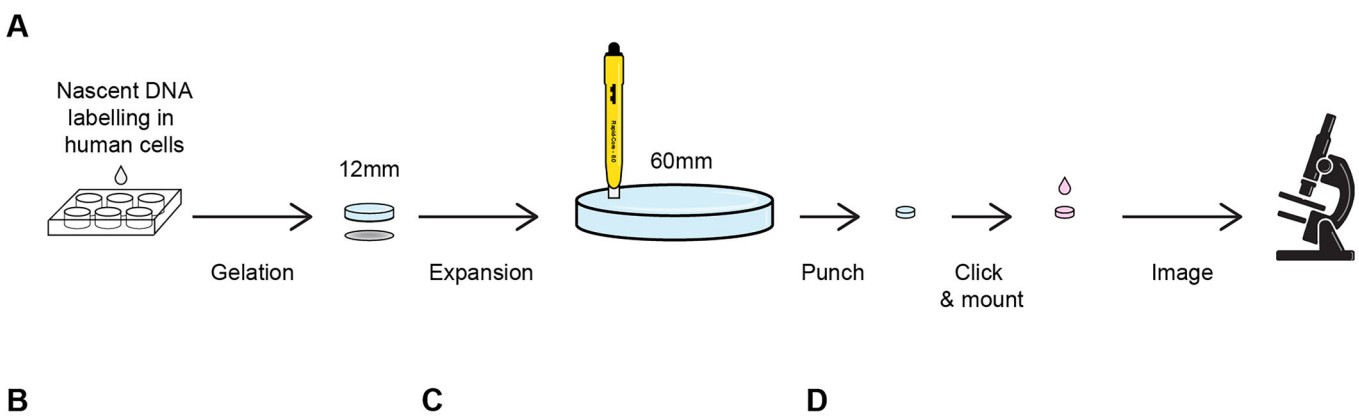

**B**

**C**

**D**

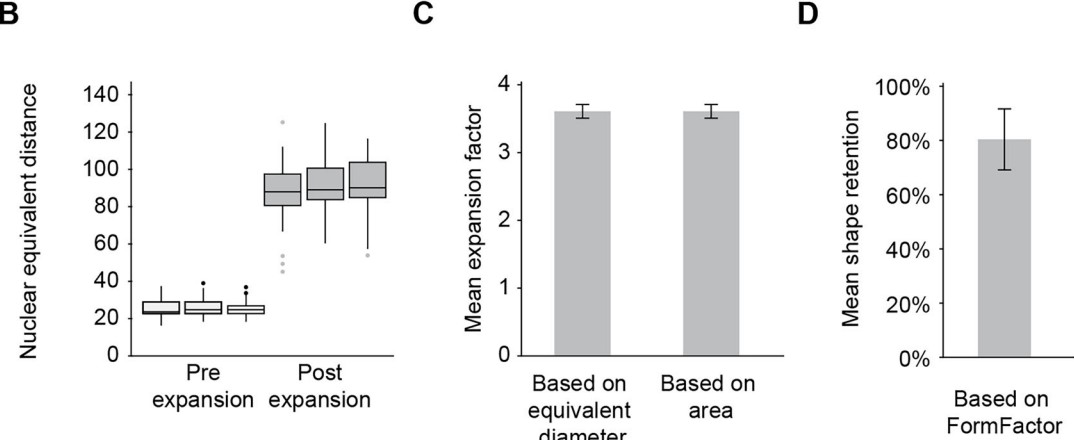

**E**

Termination events

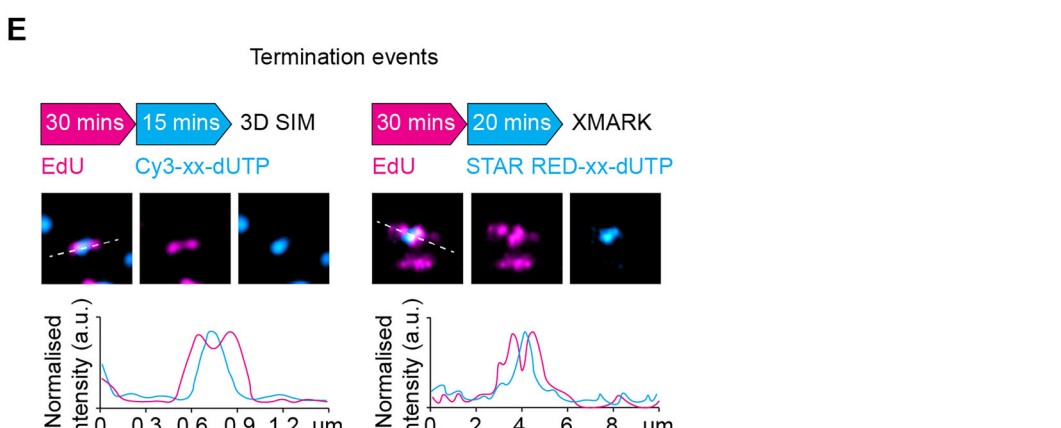

**Figure EV5. General outline and validation of XMARK protocol.**

(A) Illustration of the XMARK procedure. Cells adhered to coverslips undergo nascent DNA pulse labelling and are fixed. Gels are then formed on the coverslip, embedding the fixed cells within the gel. The formed gel is expanded, and a small section is cut, using a tissue biopsy punch tool. The cut section undergoes click chemistry and the expanded gel is mounted for confocal imaging. (B) Box plot showing the nuclear equivalent diameter of DAPI-stained single nuclei pre- and post-expansion determined by 20X wide-field microscopy and Cell Profiler software. Distribution of nuclear equivalent diameters of three independent experiments are shown ($n = 100$ cells per condition). For each box, the inner line indicates median and box limits show 25th and 75th percentiles. Whiskers extend to edge values within 1.5 times the interquartile range between 25th and 75th percentiles from the box limits. Dots represent values beyond whisker range. (C) Bar plot showing the mean expansion factor based on nuclear equivalent diametr (as depicted in B) and nuclear area (i.e. square root of the mean area of expanded nuclei divided by mean area of non-expanded nuclei). (D) Bar plot showing the mean shape retention post expansion based on FormFactor determined by 20X wide-field microscopy and Cell Profiler software. FormFactors pre- and post-expansion is calculated as $4*\pi*Area/Perimeter^2$. (E) Representative 2D maximum projection images of selected examples of termination events from a 3D-SIM and XMARK image shown in Fig. 4B, C, respectively. The line graphs show the normalised signal intensity profile (a.u.) across the termination events, illustrated by the white dashed line in the merged image.

A

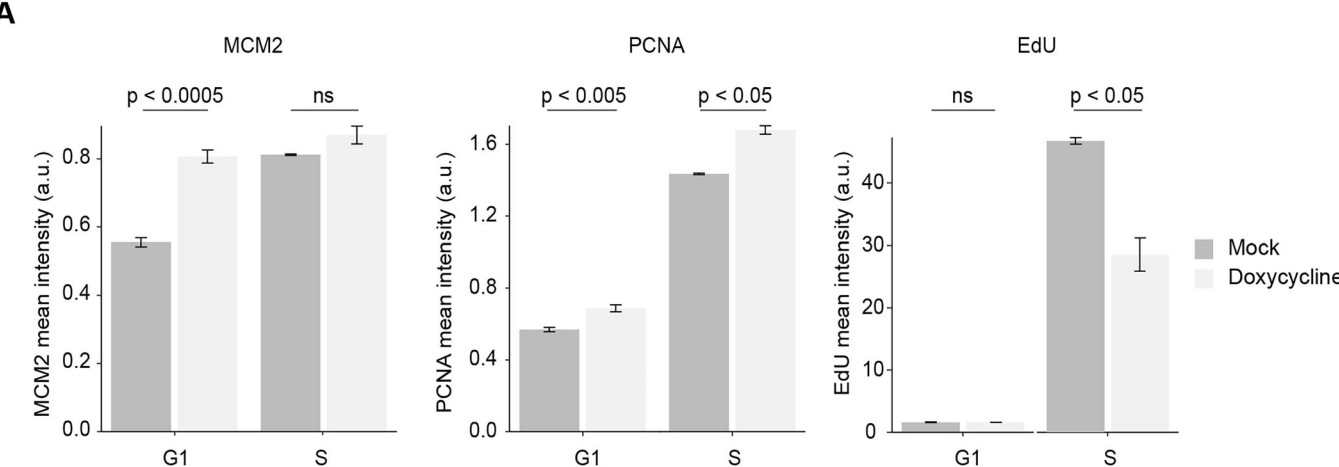

**Figure EV6.  Quantification of origin licensing and DNA synthesis in HBEC CDC6 TetON cells.**

(**A**) Bar plot showing the mean nuclear integrated intensities (a.u.) of MCM2, PCNA and EdU for asynchronous cells treated with Doxycycline for 24 h and DMSO control. G1 and S population were selected based on DAPI vs EdU. Three replicates per condition, error bars represent S.E.M (for each condition, at least 10,000 cells were analysed for G1 portion and at least 2000 cells were analysed for the S portion).

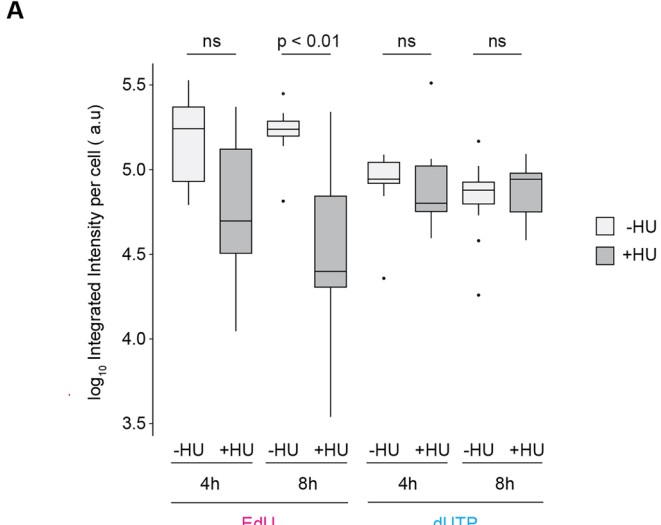

**Figure EV7. Signal intensity analysis of 3D-SPARK nano foci.**

(A) Box plot showing the nuclear integrated intensities (a.u.) of EdU and dUTP per 3D-SPARK-imaged synchronised cells with palbociclib and released for a certain amount of time prior to labelling. Number of nuclei: 10 (4 h, mock); 12 (4 h, HU); 13 (8 h, mock); 12 (8 h, HU). For each box, the inner line indicates median and box limits show 25th and 75th percentiles. Whiskers extend to edge values within 1.5 times the interquartile range between 25th and 75th percentiles from the box limits. Dots represent values beyond whisker range. P values were calculated with Mann–Whitney $U$-test and adjusted for multiple testing using Bonferroni correction. $P$: 4.8e-4 (EdU, 8 h, mock vs HU).

