## [Peer Review File · The EMBO Journal]

Spatial mapping of DNA synthesis reveals dynamics and geometry of human replication nanostructures

Michael Hawgood, Bruno Urien, Ana Agostinho, Praghadesh Thiagarajan, Giovanni Giglio, Yiqiu Yang, Xue Zhang, Gemma Quijada, Matilde Fonseca, Jiri Bartek, Hans Blom, and Bennie Lemmens

Corresponding author(s): Bennie Lemmens (bennie.lemmens@ki.se)

Review Timeline:

Submission Date:	27th Feb 25
Editorial Decision:	2nd Apr 25
Revision Received:	2nd Jul 25
Editorial Decision:	1st Sep 25
Revision Received:	5th Sep 25
Accepted:	12th Sep 25

Editor: Hartmut Vodermaier

Transaction Report:

Dr. Bennie Lemmens
Karolinska Institutet
Sweden

2nd Apr 2025

Re: EMBOJ-2025-120568
Spatial mapping of DNA synthesis reveals dynamics and geometry of human replication nanostructures

Dear Dr. Lemmens,

Thank you for submitting your manuscript on a new 3D-SPARK method for studying replication nanostructures for our consideration. We sent it to three expert referees, who have now returned the reports copied below. In light of their overall supportive comments, we would be interested in pursuing this work further for EMBO Journal publication, pending adequate revisions in response to a number of specific issues raised by the reviewers.

One key point agreed on by all referees is the inclusion of a more elaborate and ready-to-use protocol, which in fact is something we actually strongly encourage for methods/resource articles in The EMBO Journal (see <https://www.embopress.org/page/journal/14602075/authorguide#structuredmethods> for more detail). Furthermore, it would be helpful to present a bit more context about previously reported related methods, even though I would not expect a "comprehensive overview" as suggested by referee 1. There are also a few points that may be clarified with the use of additional experimentation, such as the related points by ref 1 (#5), ref 2 (#3), and ref 3's point on using synchronized cells; or ref 2 point 5 on reduced MCM phosphorylation (which ref 3 felt might be easily addressed by following MCM4 phosphorylation via western blots of nuclear extracts). On the other hand, other (technical) requests may be sufficiently answered by additional clarification, depending also on what additional validation data you may already have.

Since our single-major-revision-round policy makes it important to diligently respond to each referee point at the time of resubmission, I would encourage you to get back to me with a tentative point-by-point response and revision plan already during the early stages of the revision work, so that we could discuss how the main points might best be resolved. We would also be open to extension of the regular three-months revision period if needed; our 'scooping protection' (meaning that competing work appearing elsewhere in the meantime will not affect our considerations of your study) would of course remain valid also throughout such an extension.

Further information on preparing, formatting and uploading a revised manuscript can be found below and in our Guide to Authors. Thank you again for the opportunity to consider this work for The EMBO Journal, and I look forward to hearing from you.

With kind regards,

Hartmut

1) Every manuscript requires a Data Availability section (even if only stating that no deposited datasets are included). Primary datasets or computer code produced in the current study have to be deposited in appropriate public repositories prior to resubmission, and reviewer access details provided in case that public access is not yet allowed. Further information: [embopress.org/page/journal/14602075/authorguide#dataavailability](https://www.embopress.org/page/journal/14602075/authorguide#dataavailability)

- size of the scale bars that are mandatory for all micrograph panels
- the statistical test used to generate error bars and P-values
- the type error bars (e.g., S.E.M., S.D.)
- the number (n) and nature (biological or technical replicate) of independent experiments underlying each data point

- Figures may not include error bars for experiments with $n < 3$; scatter plots showing individual data points should be used instead.

9) To facilitate reproducibility and cross-laboratory adoption of methodologies, please structure the Materials & Methods section as outlined in our guide to authors, including a completed Reagents and Tools Table that can be downloaded from our author guidelines as well (<https://www.embopress.org/page/journal/14602075/authorguide#structuredmethods>).

10) Digital image enhancement is acceptable practice, as long as it accurately represents the original data and conforms to community standards. If a figure has been subjected to significant electronic manipulation, this must be clearly noted in the figure legend and/or the 'Materials and Methods' section. The editors reserve the right to request original versions of figures and the original images that were used to assemble the figure. Finally, we generally encourage uploading of numerical as well as gel/blot image source data; for details see: embopress.org/page/journal/14602075/authorguide#sourcedata

At EMBO Press, we ask authors to provide source data for the main manuscript figures. Our source data coordinator will contact you to discuss which figure panels we would need source data for and will also provide you with helpful tips on how to upload and organize the files.

In the interest of ensuring the conceptual advance provided by the work, we recommend submitting a revision within 3 months (1st Jul 2025). Please discuss the revision progress ahead of this time with the editor if you require more time to complete the revisions. Use the link below to submit your revision:

Link Not Available

Referee #1:

Hawgood et al present a novel dual-color replicative labeling approach, called 3D-SPARK, that combines click chemistry with labeled dUTP incorporation with a synthetic nucleoside triphosphate transporter (SNTT1). The authors validated this application with various cutting-edge microscopy techniques, including wide-field, confocal, structured illumination, and expansion microscopy. This labeling and microscopy technique is of great interest to the field, it appears straightforward, does not require sophisticated equipment, and offers flexibility in terms of fluorochrome selection. When combined with structured illumination microscopy (SIM), this labeling method enables the resolution and visualization of individual replicons and allows for the distinction of replication initiation, elongation, and termination within 3D genomic contexts. While the scientific novelty is relatively modest, the paper represents a substantial technological advancement with promising applications in understanding DNA replication dynamics and the replication stress response in a 3D genomic setting.

However, revisions are necessary to improve the clarity and comprehension of the presented data. A more detailed protocol should be included to ensure broad applicability within the research community. Additionally, the study should be contextualized within previous work in the field, providing a comprehensive overview of established methods for labeled dUTP incorporation. The authors should cite alternative dUTP incorporation approaches, including microinjections, osmotic treatment, scratch loading, and bead loading, to emphasize the added value of using SNTT1 in dual-color labeling techniques. Furthermore, a more thorough review of prior studies on the microscopic visualization of replication foci is needed. Finally, additional control experiments are required, as discussed below.

Specific comments:

- 1) The EdU labeling is interpreted as being less efficient compared to the direct incorporation of fluorophore-labeled dUTPs. However, this is likely due to the efficiency of the click chemistry reaction, and optimizing the click reaction could improve detection. Earlier techniques, such as radioactive tritium labeling and more recent nanopore sequencing-based detection of BrdU and EdU, indicate that thymidine analog incorporation in DNA occurs nearly instantaneously in mammalian cells.
- 2) The actual efficiency of incorporation efficiency of individual labeled nucleotides after two pulses has not been shown. Analyzing labeled tracks on DNA fibers is necessary to evaluate labeling efficiency and determine the ratio between the two incorporated labels. The rationale behind using mixed labeling in Figures 2A and 2B is unclear. A more rigorous validation of the double-labeling strategy would involve a chase experiment to confirm that the two colors do not overlap or colocalize after the chase.
- 3) Accurate measurements of initiation, ongoing forks, and termination events are only reliable if all cells undergo the same labeling conditions, which cannot be concluded from the current data. The authors mention that SNTT1-based labeling exhibits high variability between individual cells (lines 179-181). To mitigate potential biases, a parallel experiment using swapped EdU and Cy5-xx-dUTP pulses should be conducted to ensure robustness across different labeling schemes.
- 4) The data in Figure 1 primarily represent labeling optimization rather than results. The paper would benefit from combining Figures 1 and 2 and moving most of the panels from Figure 1 to the supplementary materials. [EDITOR's NOTE: I feel it is fine to maintain the figure structuring as is]
- 5) Fig 3B - HU treatment -increases dormant origin firing, therefore one would expect more 1-EdU - 2-dUTP structures in HU-treated cells. While Figure 6B quantifies these structures in the expansion mode, can they also be detected under unperturbed conditions in SIM mode? Does HU treatment lead to an increase in initiation events? Additionally, does HU treatment alter the mean intensity of both replicative labels? The same reasoning applies to the CDC6 overexpression experiment.
- 6) Another line of evidence suggests that early S-phase DNA replication events do not colocalize but are adjacent to transcribed regions. This observation aligns with prior studies using various techniques, including DNA combing (Demczuk et al., PLoS Biol, 2012), bubble-seq (Dijkwel et al., Mol Cell Biol, 2002), OK-seq (Petryk et al., Nat Commun), and repli-HiC (Liu et al., Science, 2024).
- 7) While this is a valuable technique that will be highly appreciated in the field, given its relatively limited scientific novelty, the authors should provide a more detailed step-by-step protocol in a supplementary file to enhance accessibility for a broader audience.

Referee #2:

In this manuscript, the authors combine labeling of nascent DNA with high resolution imaging methods to spatially examine replication in cultured cells. Many previous papers have noted changes in the spatial organization of replication during S-phase and in response to perturbations and have optimized the use of nucleoside analogues for monitoring replication. Previous papers have also utilized high resolution imaging of replication structures (see for example papers by Eli Rothenberg's laboratory). The added value in this manuscript is the careful optimization of the nascent DNA labeling methods and combination with high resolution microscopy methods. They then apply their method to several settings to demonstrate its utility. For example, they show the ability of their method to measure the frequency of initiation and termination events, the effect of reducing replication synthesis rates with hydroxyurea, the ability to combine their nascent DNA imaging with DNA damage markers, and the consequences of overexpressing CDC6 or inactivating RIF1 (two well studied regulators of replication initiation). While there isn't any major breakthroughs or new discoveries in these case studies, they do demonstrate the potential utility of the methodology. The most interesting observation is the difference in geometry of initiation events in RIF1-deficient cells. Overall, I think this manuscript fits well into the "resource" category for EMBO and offer the following recommendations for improvement prior to publishing.

1. Additional methods and analysis information is needed to be most valuable as a resource article. For example, the specific parameters of the CellProfiler and R pipelines used should be detailed better in the methods section rather than simply cite another paper. In addition, the 3D-SIM image analysis Python pipelines should be made available.
2. The authors examined how replication patterns correlate with nuclear speckles marked by SON proteins. They report that these speckles stimulate replication initiation since there is a trend of the initiation events being closer in distance to the initiation events compared to termination events. However, there is no significant difference (Figure 6F) so no conclusion can be made, and the value of this experimental result is unclear.

3. This reviewer was unclear about the limitations of the methodology in late S-phase cells. If it is limited to early to mid S-phase cells, that makes it much less useful. More information and examples about how things like initiation and termination events look in late S-phase pattern cells is important as well as how the authors chose to include or exclude cells for analyses.

4. The data analysis depends on the ability to distinguish single replication forks or replicons from clustered forks. I'm unsure how to better document this is the case, but recommend the authors fully discuss their evidence and any potential counter-arguments in the discussion.

5. CDC6 overexpression reduced the frequency of initiation events. This is counter-intuitive since CDC6 overexpression should generate more licensed origins. To explain the result, the authors cite a yeast paper that showed the initiation kinase CDC7 is sequestered when there is a lot of unfired origins preventing it from targeting substrates other than the MCM complex (Seoane and Morgan, 2017). However, that results does not show that there would be less MCM phosphorylation and therefore less initiation when CDC6 is overexpressed. Further explanation and validation that there are truly less initiation events in early S-phase cells is needed.

Referee #3:

DNA replication foci are still elusive objects observed in the nuclei of every cell and are presumably involved in the arrangements of nuclear domains and proteins. They have been the subjects of several explanations over the years. However, they still deserve new approaches to deciphering their structure and relationship with the genome's dynamic organisation for DNA replication. This manuscript provides a new analysis of the DNA replication foci using an elaborated technical approach (3D-SPARK).

Overall, I found this manuscript both very interesting and especially well-written. It provides precise explanations of the method used and the expected results.

My main comments are as follows:

The introduction is clear and summarises the interest of this analysis well. Results:

In Figure 1B, showing an example of labelling with the second approach used by the authors would be nice.

Relationship between the size of labelled DNA and the number of base pairs: A 15-min pulse corresponds to around 30 Kb length of synthesised DNA at the replication forks during the elongation process. This will correspond to a 30 Kb length at a DNA replication origin because the two forks in opposite directions will not be discriminated. In the manuscript, the length of labelled DNA is estimated to be 16 Kbp, based on a correspondence between the DNA length in nm and a published value of 2.59 Kbp/min in a different cell. This value might be incorrect in the cells and conditions used in this manuscript. First, using modified dNTPs will likely slow down the replication forks, and differently according to the modified dNTP used. Therefore, I suggest the authors accurately measure the replication speed in their conditions. It will improve the quantification aspects of this manuscript. The optimisation of the nascent DNA labelling part is very instructive and comprehensibly commented.

In the experiment using HU, it would have been nice to make this analysis on synchronised cells. The HU treatment of a synchronised cell population at different times should provide nice data.

The paragraph concerning using 3D-SPARKS in analysing perturbations in DNA replication is also quite convincing.

In conclusion, it is an excellent analysis that provides several new improvements to the methods used to analyse in-vivo DNA replication in individual cells, both at the qualitative and quantitative levels. I anticipate possible criticisms about the novelty in our knowledge of the nature of DNA replication foci. However, I believe this manuscript will be a helpful technical contribution to this still unclear field.

Response to the reviewers concerning revised manuscript EMBOJ-2025-120568

We thank the reviewers for their thorough evaluation and valuable feedback. We are pleased that all three referees recognized the novelty of our experimental approach and broad utility of the 3D-SPARK technology for cell biology/genome research.

To facilitate the implementation of 3D-SPARK approach by our colleagues in the field, and comply to the request of all three referees, we now added a detailed wet-lab protocol and a link to our annotated 3D image analysis pipelines. This step-by-step protocol include practical instructions for dual-colour nascent DNA labelling in human cells and sample prepping for 3D-SIM and ExM.

We also performed additional control experiments and provide more context related to previous nascent DNA labelling approaches and related super-resolution microscopy studies. We provide new time-resolved 3D-SIM measurements using synchronized cells and discuss the challenges to study DNA replication dynamics in late-S phase cells. The revised manuscript includes a new main figure and nine additional data panels (including single-cell and nanoscale analyses) that clarify the methodology and further solidify our observations upon HU treatment and CDC6 overexpression. We are grateful for your directions and the recommendations, which we believe significantly strengthen the manuscript.

Below we will address all individual points raised by the reviewers.

Point-by-point response:**REVIEWERS' COMMENTS**

Referee #1:

Hawgood et al present a novel dual-color replicative labeling approach, called 3D-SPARK, that combines click chemistry with labeled dUTP incorporation with a synthetic nucleoside triphosphate transporter (SNTT1). The authors validated this application with various cutting-edge microscopy techniques, including wide-field, confocal, structured illumination, and expansion microscopy. This labeling and microscopy technique is of great interest to the field, it appears straightforward, does not require sophisticated equipment, and offers flexibility in terms of fluorochrome selection. When combined with structured illumination microscopy (SIM), this labeling method enables the resolution and visualization of individual replicons and allows for the distinction of replication initiation, elongation, and termination within 3D genomic contexts. While the scientific novelty is relatively modest, the paper represents a substantial technological advancement with promising applications in understanding DNA replication dynamics and the replication stress response in a 3D genomic setting. However, revisions are necessary to improve the clarity and comprehension of the presented data. A more detailed protocol should be included to ensure broad applicability within the research community. Additionally, the study should be contextualized within previous work in the field, providing a comprehensive overview of established methods for labeled dUTP incorporation. The authors should cite alternative dUTP incorporation approaches, including microinjections, osmotic treatment, scratch loading, and bead loading, to emphasize the added value of using SNTT1 in dual-color labeling techniques.

Furthermore, a more thorough review of prior studies on the microscopic visualization of replication foci is needed. Finally, additional control experiments are required, as discussed below.

52 Thank you for your positive comments and for helping us improve our manuscript. The manuscript
53 now includes a detailed step-by-step protocol (snapshot below and supplementary file
54 **Protocol_EV1**) and a GitHub link to our custom image analysis scripts.
55

56
57
58 We also added an extra introductory and discussion paragraph about nascent DNA labelling
59 methods and previous nanoscale replication studies, focussing on the added value of SNTT1-based
60 methods compared to other nucleoside delivery approaches (page 2 and 11 of the revised
61 manuscript).
62

63 Specific comments:

64
65 1) The EdU labeling is interpreted as being less efficient compared to the direct incorporation of
66 fluorophore-labeled dUTPs. However, this is likely due to the efficiency of the click chemistry reaction,
67 and optimizing the click reaction could improve detection. Earlier techniques, such as radioactive
68 tritium labeling and more recent nanopore sequencing-based detection of BrdU and EdU, indicate that
69 thymidine analog incorporation in DNA occurs nearly instantaneously in mammalian cells.
70

71 We agree that thymidine analogues such as EdU can enter the cell instantaneously and incorporate
72 into the genome within minutes, (e.g. we have observed numerous incorporated EdU molecules within
73 3 minutes using highly sensitive Single Molecule Localization Microscopy methods – unpublished data).
74 However, to incorporate EdU into DNA, it must first be phosphorylated inside the cell through a series
75 of enzymatic steps involving thymidine kinase, pyrimidine nucleoside monophosphate kinase and
76 nucleoside diphosphate kinase to form the active triphosphate form EdUTP suitable for DNA synthesis.
77 While EdU detection relies on efficient click-chemistry, which indeed can be optimized per application,
78 we wish the emphasise that the biological activation of EdU to EdUTP remains an inherent limiting
79 factor for short pulse timing. To further illustrate this phenomenon, we perform a time-series EdU
80 pulse experiment (akin to Figure 2B of Pereira et al. Oncotarget. 2017,
81 <https://doi.org/10.18632/oncotarget.17121>) and quantified integrated EdU intensities for each EdU
82 pulse duration (ranging from 5 min to 2 hours) using identical click-chemistry conditions. For pulses
83 longer than 1 hour, EdU intensities correlate linearly with EdU pulse length (Pereira et al. 2017).
84 However, for pulses shorter than 1 hour, the relationship becomes non-linear and EdU intensities
85 diminish relative to their respective pulse length:

These findings confirm the data presented in Figure 1F and argue that *in situ* EdU modification/incorporation is restrictive during short pulses. While we agree (some) EdU gets incorporated within minutes, we believe the quantitative data presented here can be useful for the community when planning nucleoside-based imaging studies. The time-series EdU experiment is included as **Figure EV1D** in the revised manuscript.

2) The actual efficiency of incorporation efficiency of individual labeled nucleotides after two pulses has not been shown. Analyzing labeled tracks on DNA fibers is necessary to evaluate labeling efficiency and determine the ratio between the two incorporated labels. The rationale behind using mixed labeling in Figures 2A and 2B is unclear. A more rigorous validation of the double-labeling strategy would involve a chase experiment to confirm that the two colors do not overlap or colocalize after the chase.

We agree that direct comparisons of EdU vs Cy3-dUTP using DNA fibres would be insightful, however, the selected labels (while well-suited for in situ super-resolution imaging) are poorly suited for DNA fibre analysis. The harsh conditions and detergents needed for cell lysis and fibre spreading are detrimental for effective EdU click chemistry and compromise the signal of cyanine dyes. The reciprocal nearest neighbour analysis depicted in Figure 2F indicates that the two sequential labels occur at similar densities, suggesting low levels of α -specific staining or inefficient EdU/dUTP labelling events. Following the referee's recommendation, we performed additional pulse-chase experiments to validate the colocalization signals observed using dual-colour 3D-SPARK. Simultaneous addition of Cy3-xx-dUTP and Cy5-xx-dUTP resulted in near-identical, highly overlapping foci patterns, confirming signal specificity and our earlier observations in Figure 2B. In contrast, introducing a 30-minute chase between Cy3-xx-dUTP and Cy5-xx-dUTP generated distinct foci patterns with limited overlap in Cy3/Cy5 signal intensities. Analogous experiments based on our *standard pulsing scheme*, revealed that a 30-minute chase between EdU and Cy3-xx-dUTP labels diminished the volume overlap and significantly increased the centroid distances between associated EdU and Cy3 signals (see below). These extra controls are now included as **Figure EV2A-B** and discussed in the main text.

3) Accurate measurements of initiation, ongoing forks, and termination events are only reliable if all cells undergo the same labeling conditions, which cannot be concluded from the current data. The authors mention that SNTT1-based labeling exhibits high variability between individual cells (lines 179-181). To mitigate potential biases, a parallel experiment using swapped EdU and Cy5-xx-dUTP pulses should be conducted to ensure robustness across different labeling schemes.

135 While labelling variability is indeed larger for the SNTT1-
 136 based method compared to EdU, we like to emphasize that
 137 both methods consistently label all S-phase cells under the
 138 conditions tested. We performed “dye swap” studies in
 139 which we labelled the cells with EdU *first* + Cy3-xx-dUTP
 140 *second* versus Cy3-xx-dUTP *first* + EdU *second*, and compared
 141 initiation/ongoing/termination events ratios by 3D-SPARK
 142 (right figure). These “dye swap” studies confirmed that the
 143 majority of dual-colour events are “ongoing events” and are
 144 consistent with the general 1:8:1 ratio for
 145 initiation/ongoing/termination events in human RPE1 cells.
 146 We noted a significant reduction in initiation events when the
 147 Cy3-xx-dUTP label was added first, possibly reflecting altered
 148 replication kinetics/fork symmetry by the bulky dye.
 149 Together, these results validate our 3D-SPARK approach and
 150 image analysis pipelines and support our recommendation to
 151 use the smallest nucleoside first to promote fork
 152 processivity. This graph is included as **Figure EV4B** and
 153 discussed in the main text.

154
 155 4) The data in Figure 1 primarily represent labeling optimization rather than results. The paper would
 156 benefit from combining Figures 1 and 2 and moving most of the panels from Figure 1 to the
 157 supplementary materials. [EDITOR's NOTE: I feel it is fine to maintain the figure structuring as is]

158
 159 We agree with the editor and favour to keep Figure 1 in main text, mainly because it provides the
 160 community several practical and conceptual considerations when studying replication dynamics.

161
 162 5) Fig 3B - HU treatment -increases dormant origin firing, therefore one would expect more 1-EdU -
 163 2-dUTP structures in HU-treated cells. While Figure 6B quantifies these structures in the expansion
 164 mode, can they also be detected under unperturbed conditions in SIM mode? Does HU treatment
 165 lead to an increase in initiation events? Additionally, does HU treatment alter the mean intensity of
 166 both replicative labels? The same reasoning applies to the CDC6 overexpression experiment.

167
 168 We apologize for the confusion on Figure 6B; HU treatment indeed leads to an increase in initiation
 169 events. We initially observed the increase in initiation events in non-synchronized RPE1 cells using 3D-
 170 SIM (figure 6B) and now independently confirmed it in synchronized cells (**Figure 7D in the revised**
 171 **manuscript**, see comments to referee #3).

We also include mean intensities for each replicative label, demonstrating that HU reduces EdU but not dUTP intensities (**Figure EV7**). These findings fit the notion that DNA synthesis is rapidly restored upon HU release (Petermann E *et al.* Mol Cell 2010 and our own unpublished data), given that HU is present during the entire EdU pulse, but removed during the Cy3-xx-dUTP transfection. We find CDC6 overexpression to affect both replicative labels, in line with stable protein expression, and provide additional imaging data to support the observed reduction in initiation events (**Figure EV6**).

184 6) Another line of evidence suggests that early S-phase DNA
 185 replication events do not colocalize but are adjacent to transcribed regions. This observation aligns
 with prior studies using various techniques, including DNA combing (Demczuk et al., PLoS Biol, 2012),

186 bubble-seq (Dijkwel et al., Mol Cell Biol, 2002), OK-seq (Petryk et al., Nat Commun), and repli-HiC (Liu
187 et al., Science, 2024).

188

189 Thank you for the additional insights and scientific context. We now included these literature
190 references to the revised text (page 9) and also ran additional 3D analysis of SON speckles and
191 replication nanostructures. We quantified the fraction of dUTP signals overlapping with SON or EdU
192 foci (**Figure 6G in the revised manuscript**), and subsequently, for the dUTP signals that overlap with
193 either SON or EdU foci, we plotted the proportion of volume overlap (**Figure 6H in the revised
194 manuscript**). Both metrics reveal significant exclusion of nascent DNA from SON speckles compared
195 to EdU foci:

196

197

198 7) While this is a valuable technique that will be highly appreciated in the field, given its relatively
199 limited scientific novelty, the authors should provide a more detailed step-by-step protocol in a
200 supplementary file to enhance accessibility for a broader audience.

201

202 We now include a step-by-step protocol in supplementary file **Protocol_EV1**).

203

204 Referee #2:

205

206 In this manuscript, the authors combine labeling of nascent DNA with high resolution imaging
207 methods to spatially examine replication in cultured cells. Many previous papers have noted changes
208 in the spatial organization of replication during S-phase and in response to perturbations and have
209 optimized the use of nucleoside analogues for monitoring replication. Previous papers have also
210 utilized high resolution imaging of replication structures (see for example papers by Eli Rothenberg's
211 laboratory). The added value in this manuscript is the careful optimization of the nascent DNA
212 labeling methods and combination with high resolution microscopy methods. They then apply their
213 method to several settings to demonstrate its utility. For example, they show the ability of their
214 method to measure the frequency of initiation and termination events, the effect of reducing
215 replication synthesis rates with hydroxyurea, the ability to combine their nascent DNA imaging with
216 DNA damage markers, and the consequences of overexpressing CDC6 or inactivating RIF1 (two well
217 studied regulators of replication initiation). While there isn't any major breakthroughs or new
218 discoveries in these case studies, they do demonstrate the potential utility of the methodology. The
219 most interesting observation is the difference in geometry of initiation events in RIF1-deficient cells.
220 Overall, I think this manuscript fits well into the "resource" category for EMBO and offer the
221 following recommendations for improvement prior to publishing.

222

223 Thank you for your careful review and shared enthusiasm for the 3D-SPARK methodology.

224

225 1. Additional methods and analysis information is needed to be most valuable as a resource article.
226 For example, the specific parameters of the CellProfiler and R pipelines used should be detailed
227 better in the methods section rather than simply cite another paper. In addition, the 3D-SIM image
228 analysis Python pipelines should be made available.

229

230 We now include a detailed protocol (supplementary file **Protocol_EV1**) and annotated GitHub link in
231 the method section describing the used parameters of our custom CellProfiler, Python and R Studio
232 pipelines.

233 2. The authors examined how replication patterns correlate with nuclear speckles marked by SON
234 proteins. They report that these speckles stimulate replication initiation since there is a trend of the
235 initiation events being closer in distance to the initiation events compared to termination events.
236 However, there is no significant difference (Figure 6F) so no conclusion can be made, and the value of
237 this experimental result is unclear.
238

239 The revised manuscript now includes additional 3D image analyses showing significant exclusion of
240 nascent DNA foci from SON speckles (Figure 6G and 6H). While the initial data only showed a trend,
241 these data support the compatibility of 3D-SPARK with IF biomarkers and together provide proof-of-
242 principle of direct spatial correlation between DNA replication dynamics and nuclear organelles. We
243 hope this inspires the community to use 3D-SPARK to address the myriad spatial relationships within
244 the human nucleus.
245

246 3. This reviewer was unclear about the limitations of the methodology in late S-phase cells. If it is
247 limited to early to mid S-phase cells, that makes it much less useful. More information and examples
248 about how things like initiation and termination events look in late S-phase pattern cells is important
249 as well as how the authors chose to include or exclude cells for analyses.
250

251 This is indeed an important topic and resolving replication events during mid/late S phase is
252 particularly challenging due to clustering of replication foci. We now added new experimental data
253 directly addressing this issue by analysing replication nanofoci at defined S-phase timings (new
254 Figure 7, see comments to referee #3). While mid/late S phase cells indeed showed more foci in
255 crowded regions, our 3D-SPARK pipelines remained capable of extracting hundreds of discrete dual-
256 colour events per nucleus and detecting initiation and termination levels upon drugs exposure.
257 These data sets reveal differential responses to HU in early *versus* mid/late S phase and provide
258 additional data-driven considerations when studying replication foci throughout S-phase.
259

260 4. The data analysis depends on the ability to distinguish single replication forks or replicons from
261 clustered forks. I'm unsure how to better document this is the case, but recommend the authors fully
262 discuss their evidence and any potential counter-arguments in the discussion.
263

264 We now elaborate on the detection of single replication forks in the discussion section and propose
265 future directions to detect replication forks by STORM and MINFLUX. We are currently using the
266 latest single-molecule localisation microscopy (SMLM) setups to detect what is 'inside' a replication
267 focus and identified string/loop-like nascent DNA structures. While in-depth SMLM studies will
268 provide additional molecular and structural insights into human replication forks, we consider these
269 studies beyond the scope of this study focussed on replication dynamics using dual-pulse labelling.
270 In the revised manuscript we discuss how SMLM can help resolving dense foci clusters (e.g. in late S-
271 phase cells) but also emphasise that numerous valuable insights can be gained from analysing DNA
272 replication at ~100 nanometre spatial resolution and directly relating biomarkers with two nascent
273 DNA markers in the same human cell – a feat well performed by 3D-SIM and ExM.
274

275 5. CDC6 overexpression reduced the frequency of initiation events. This is counter-intuitive since
276 CDC6 overexpression should generate more licensed origins. To explain the result, the authors cite a
277 yeast paper that showed the initiation kinase CDC7 is sequestered when there is a lot of unfired
278 origins preventing it from targeting substrates other than the MCM complex (Seoane and Morgan,
279 2017). However, that results does not show that there would be less MCM phosphorylation and
280 therefore less initiation when CDC6 is overexpressed. Further explanation and validation that there
281 are truly less initiation events in early S-phase cells is needed.
282

283 We agree that reduced DNA synthesis upon overexpression of licencing factors might feel counter-
 284 intuitive, but the Meyer lab has revealed that core licencing factors such as CDT1 directly inhibit DNA
 285 synthesis in early S phase (Ratnayake N et al. Mol Cell. 2023 [10.1016/j.molcel.2022.12.004](https://doi.org/10.1016/j.molcel.2022.12.004)) and the
 286 initial description of the CDC6-TetON HBEC cells by the Gorgoulis lab also demonstrated a stark drop
 287 in DNA synthesis within 12 hours of CDC6 overexpression (Zampetidis et al. Mol Cell. 2021
 288 <https://doi.org/10.1016/j.molcel.2021.10.017>). To complement our 3D SPARK data, we performed
 289 additional high-content imaging experiments upon CDC6 overexpression and simultaneously assessed
 290 origin licensing and DNA synthesis in single cells:

291 Our data indicates that CDC6 overexpression impairs EdU incorporation while causing an increase in
 292 chromatin-bound MCM and PCNA, confirming that over licensing does not translate to increased DNA
 293 synthesis but instead hampers efficient genome replication. These data are included as **Figure EV6**
 294 and support the reduction in initiation events detected by 3D-SPARK (our study) as well as the
 295 replication stress phenotypes observed by Zampetidis and colleagues.
 296
 297

298 Referee #3:

299
 300 DNA replication foci are still elusive objects observed in the nuclei of every cell and are presumably
 301 involved in the arrangements of nuclear domains and proteins. They have been the subjects of
 302 several explanations over the years. However, they still deserve new approaches to deciphering their
 303 structure and relationship with the genome's dynamic organisation for DNA replication. This
 304 manuscript provides a new analysis of the DNA replication foci using an elaborated technical
 305 approach (3D-SPARK).
 306 Overall, I found this manuscript both very interesting and especially well-written. It provides precise
 307 explanations of the method used and the expected results.
 308

309 Thank you for your positive comments and valuable insights regarding the context of our work.

310
 311 My main comments are as follows:

312
 313 The introduction is clear and summarises the interest of this analysis well. Results:

314 In Figure 1B, showing an example of labelling with the second approach used by the authors would
 315 be nice.

316 We now included an example image of F-ara-EdU (100uM, 4 hours) as a positive control (**Figure**
 317 **EV1A**).

318

319
320
321
322
323
324
325
326
327
328
329
330
331
332
333
334
335
336
337
338
339
340
341
342
343
344
345
346
347
348
349
350
351
352
353
354
355
356
357
358
359
360
361
362
363
364
365
366
367
368
369

Relationship between the size of labelled DNA and the number of base pairs: A 15-min pulse corresponds to around 30 Kbp length of synthesised DNA at the replication forks during the elongation process. This will correspond to a 30 Kbp length at a DNA replication origin because the two forks in opposite directions will not be discriminated. In the manuscript, the length of labelled DNA is estimated to be 16 Kbp, based on a correspondence between the DNA length in nm and a published value of 2.59 Kbp/min in a different cell. This value might be incorrect in the cells and conditions used in this manuscript. First, using modified dNTPs will likely slow down the replication forks, and differently according to the modified dNTP used. Therefore, I suggest the authors accurately measure the replication speed in their conditions. It will improve the quantification aspects of this manuscript. The optimisation of the nascent DNA labelling part is very instructive and comprehensively commented.

While we agree that direct comparisons of fork speeds of EdU versus Cy3-dUTP tracks using DNA fibers would be informative, the harsh conditions and detergents needed for DNA fibre assays prevent efficient EdU click chemistry and weaken cyanine dyes. We selected these labels because they are well-suited for multiple *in situ* super-resolution imaging modalities, but unfortunately were unable to obtain EdU+Cy3-dUTP positive fibres. We reckon the best way to accurately compare fork velocities upon the different nascent DNA labels would be to use the latest nanopore sequencing technologies (Theulot, B et al. Nat Commun 2022 <https://doi.org/10.1038/s41467-022-31012-0>), but this would require extensive optimisation, modelling and validation of unique nanopore signatures of both EdU and Cy3-dUTP, which we believe is beyond the scope of this study. While we fully agree that the 16 Kbp estimate relies on spreading efficacy, the basic goal of our fibre experiment was to determine whether the SNTT-based method created tracks smaller than the average inter-origin distance (100 Kbp), which remains the case even if our estimate would be 5-fold higher. Given the rapid developments in nanopore sequencing and AI-driven pattern recognition we hope that we can relate 3D spatial data directly to fork velocities in a not-too distant future.

In the experiment using HU, it would have been nice to make this analysis on synchronised cells. The HU treatment of a synchronised cell population at different times should provide nice data. The paragraph concerning using 3D-SPARKS in analysing perturbations in DNA replication is also quite convincing.

We thank the reviewer for this excellent suggestion which spurred us to study 3D DNA replication dynamics and HU drug responses in synchronised S-phase cells.

We synchronised RPE1 cells using low-dose palbociclib and studied replication initiation/termination rates in early and late S-phase cells using 3D-SIM (Figure 7A – here on the right). These data confirmed the typical spatiotemporal foci patterns in S-phase (used to classify S phase cells in non-synchronised cells) including the clustering of late-replication events (Figure 7B-C). Notably, this analyses also revealed nanoscale differences in replication stress response in early versus late S-phase,

370 including a relative increase in replication initiation in HU-treated late S-phase cells (**Figure 7D**). The
371 latter is in line with increasing CDK activity throughout S-phase (Lemmens et al. 2018,
372 <https://doi.org/10.1016/j.molcel.2018.05.026>) and CyclinA-Cdk1 promoting dormant origin firing
373 (Katsuno Y et al. PNAS 2009 [10.1073/pnas.0809350106](https://doi.org/10.1073/pnas.0809350106)). We included these findings in a new main
374 figure (**Figure 7**) of the revised manuscript.
375

376 In conclusion, it is an excellent analysis that provides several new improvements to the methods
377 used to analyse in-vivo DNA replication in individual cells, both at the qualitative and quantitative
378 levels. I anticipate possible criticisms about the novelty in our knowledge of the nature of DNA
379 replication foci. However, I believe this manuscript will be a helpful technical contribution to this still
380 unclear field.

Dr. Bennie Lemmens
Karolinska Institutet
Department of Medical Biochemistry and Biophysics
Sweden

1st Sep 2025

Re: EMBOJ-2025-120568R
Spatial mapping of DNA synthesis reveals dynamics and geometry of human replication nanostructures

Dear Dr. Lemmens,

Thank you for submitting your revised manuscript to The EMBO Journal. It has now been re-reviewed by original referees 1 and 2, who were both fully satisfied with the revisions. We shall therefore be happy to proceed with acceptance and production of the study for publication, as soon as the following editorial issues have also been adequately dealt with:

- Our routine pre-acceptance image checks indicate that an apparently empty black panel is shown for the "EdU" inset magnification at the top of Figure 3C, without visible background signals even upon routine setting adjustments with image-processing software. Could you please carefully double-check whether the inset crop is truly derived from the larger image field on the left of the figure, and confirm that there was indeed no detectable signal of any kind here? If so, this may warrant a brief disclaimer/statement in the respective figure legend.
- As we are switching from a free-text author contribution statement towards a more formal statement based on Contributor Role Taxonomy (CRediT) terms, please remove the present Author Contribution section and instead specify each author's contribution(s) directly in the Author Information page of our submission system during upload of the final manuscript. See <https://casrai.org/credit/> for more information.
- Please correct the likely erroneous reference to "Figure 7D-7E" on page 10 into Figure 8D-8E, which are currently lacking a call-out.
- Please organize the Source Data file so that there is one zipped folder for each main figure (containing individual files for each of the panels of the respective figure, with clear labelling for figure and figure panel numbers). Only the Source Data for the Expanded View Figures should be combined in a shared ZIP folder, again being internally organized into folders & files according to EV Figure number/panel.
- Please provide suggestions for a short 'blurb' text prefacing and summing up the conceptual aspect of the study in two sentences (max. 250 characters), followed by 3-5 one-sentence 'bullet points' with brief factual statements of key results of the paper; they will form the basis of an editor-written 'Synopsis' accompanying the online version of the article. Please also upload a synopsis image, which can be used as a "visual title" for the synopsis section of your paper. The image should be in PNG or JPG format, and please make sure that it remains in the modest dimensions of (exactly) 550 pixels wide and 300-600 pixels high.
- Finally, I think the best way of presenting the helpful and nicely illustrated step-by-step 3D-SPARK protocol within our publication format shall be by including it as an "Appendix" PDF. For this, please pre-face the protocol with a title page stating "Appendix for [article title] by {authors}" and "Table of Contents: Appendix Protocol". Furthermore, please make sure to reference the "Appendix Protocol" (at least once) from the main text, e.g. on appropriate occasions in Results/Discussion and Methods sections.

I am returning the manuscript to you for a final round of minor revision, to allow you to make these modifications and upload the revised files. Once we will have received them, we should be ready to swiftly proceed with formal acceptance and production of the manuscript.

With kind regards,

Hartmut

*** PLEASE NOTE: All revised manuscripts are subject to initial checks for completeness and adherence to our formatting guidelines. Revisions may be returned to the authors and delayed in their editorial re-evaluation if they fail to comply to the following requirements (see also our Guide to Authors for further information):

9) To facilitate reproducibility and cross-laboratory adoption of methodologies, please structure the Materials & Methods section as outlined in our guide to authors, including a completed Reagents and Tools Table that can be downloaded from our author guidelines as well (<https://www.embopress.org/page/journal/14602075/authorguide#structuredmethods>).

10) Digital image enhancement is acceptable practice, as long as it accurately represents the original data and conforms to community standards. If a figure has been subjected to significant electronic manipulation, this must be clearly noted in the figure legend and/or the 'Materials and Methods' section. The editors reserve the right to request original versions of figures and the original images that were used to assemble the figure. Finally, we generally encourage uploading of numerical as well as gel/blot image source data; for details see: embopress.org/page/journal/14602075/authorguide#sourcedata

In the interest of ensuring the conceptual advance provided by the work, we recommend submitting a revision within 3 months (30th Nov 2025). Please discuss the revision progress ahead of this time with the editor if you require more time to complete the revisions. Use the link below to submit your revision:

Link Not Available

Referee #1:

The authors have thoroughly addressed all referee comments, and the inclusion of detailed protocols substantially enhances the manuscript, ensuring its broader significance and impact within the field

Referee #2:

The authors have satisfactorily answered my questions. I have no further comments.